# Covalent PARylation of DNA base excision repair proteins regulates DNA demethylation

Simon D. Schwarz [1,6], Jianming Xu[1,2,6], Kapila Gunasekera [2,3],
David Schürmann [1], Cathrine B. Vågbø[4], Elena Ferrari[2], Geir Slupphaug [4],
Michael O. Hottiger [2], Primo Schär [1] ✉ & Roland Steinacher [1,5] ✉

The intracellular ATP-ribosyltransferases PARP1 and PARP2, contribute to DNA base excision repair (BER) and DNA demethylation and have been implicated in epigenetic programming in early mammalian development. Recently, proteomic analyses identified BER proteins to be covalently poly-ADP-ribosylated by PARPs. The role of this posttranslational modification in the BER process is unknown. Here, we show that PARP1 senses AP-sites and SSBs generated during TET-TDG mediated active DNA demethylation and covalently attaches PAR to each BER protein engaged. Covalent PARylation dissociates BER proteins from DNA, which accelerates the completion of the repair process. Consistently, inhibition of PARylation in mESC resulted both in reduced locus-specific TET-TDG-targeted DNA demethylation, and in reduced general repair of random DNA damage. Our findings establish a critical function of covalent protein PARylation in coordinating molecular processes associated with dynamic DNA methylation.

Active BER-mediated DNA demethylation occurs via stepwise oxidation of 5-methylcytosine (mC) to 5-formylcytosine (fC) and 5-carboxylcytosine (caC) by ten-eleven translocation (TET) proteins. The thymine DNA glycosylase (TDG) then recognizes and excises fC and caC, thereby engaging BER to restore an unmodified C[1]. In naïve mouse embryonic stem cells (mESC), TET activity continuously generates fC and caC with a steady state level of about 60'000 per genome[2]. TDG converts these fC and caC bases to transient abasic sites (AP-sites) and DNA single strand-breaks (SSBs). PARP1 and PARP2 bind DNA SSBs, which stimulates their poly(ADP-ribosylation) activity and covalent attachment of poly(ADP) residues (PARylation) to themselves and nearby proteins[3,4]. In BER, non-covalent interactions of XRCC1 and its partners LIG3, APE1, and POLβ with PAR were shown to facilitate their recruitment to SSBs and to accelerate repair[3–7]. This implies that mESC depend on a high PARylation potential to be able to

deal with the relatively high levels of SSBs generated by DNA demethylation[8]. In line with this are previous findings that mESC are sensitive to PARP1 inhibition and that this sensitivity is rescued by inactivation of TDG[2,9]. Proteomic data identified BER proteins (i.e., TDG, APE1, POLβ, XRCC1 and LIG3) amongst covalently PARylated proteins in unchallenged pluripotent mESC[10], and H2O2 treated HeLa cells[11]. The functional consequences of this posttranslational modification on BER are not understood.

To address the molecular function of covalent BER protein PARylation, we reconstituted TDG-BER dependent DNA demethylation in vitro with purified proteins under conditions facilitating PARylation, validated the findings in mESC and studied the functional consequences of PARylation-site-mutated BER scaffold protein XRCC1 in U2OS cells. We found that PARP1 senses the AP-site and SSB BER repair intermediates. This activates auto-PARylation of PARP1 as expected

[1]Department of Biomedicine, University of Basel, Basel, Switzerland. [2]Department of Molecular Mechanisms of Disease, University of Zurich, Zurich, Switzerland. [3]Department of Chemistry, Biochemistry and Pharmaceutical Sciences, University of Bern, Bern, Switzerland. [4]Proteomics and Modomics Experimental Core Facility (PROMEC), Norwegian University of Science and Technology and St. Olavs Hospital, Trondheim, Norway. [5]Institute of Molecular Health Sciences, ETH Zurich, Zurich, Switzerland. [6]These authors contributed equally: Simon D. Schwarz, Jianming Xu. ✉e-mail: Primo.Schaer@unibas.ch; Roland.Steinacher@biol.ethz.ch

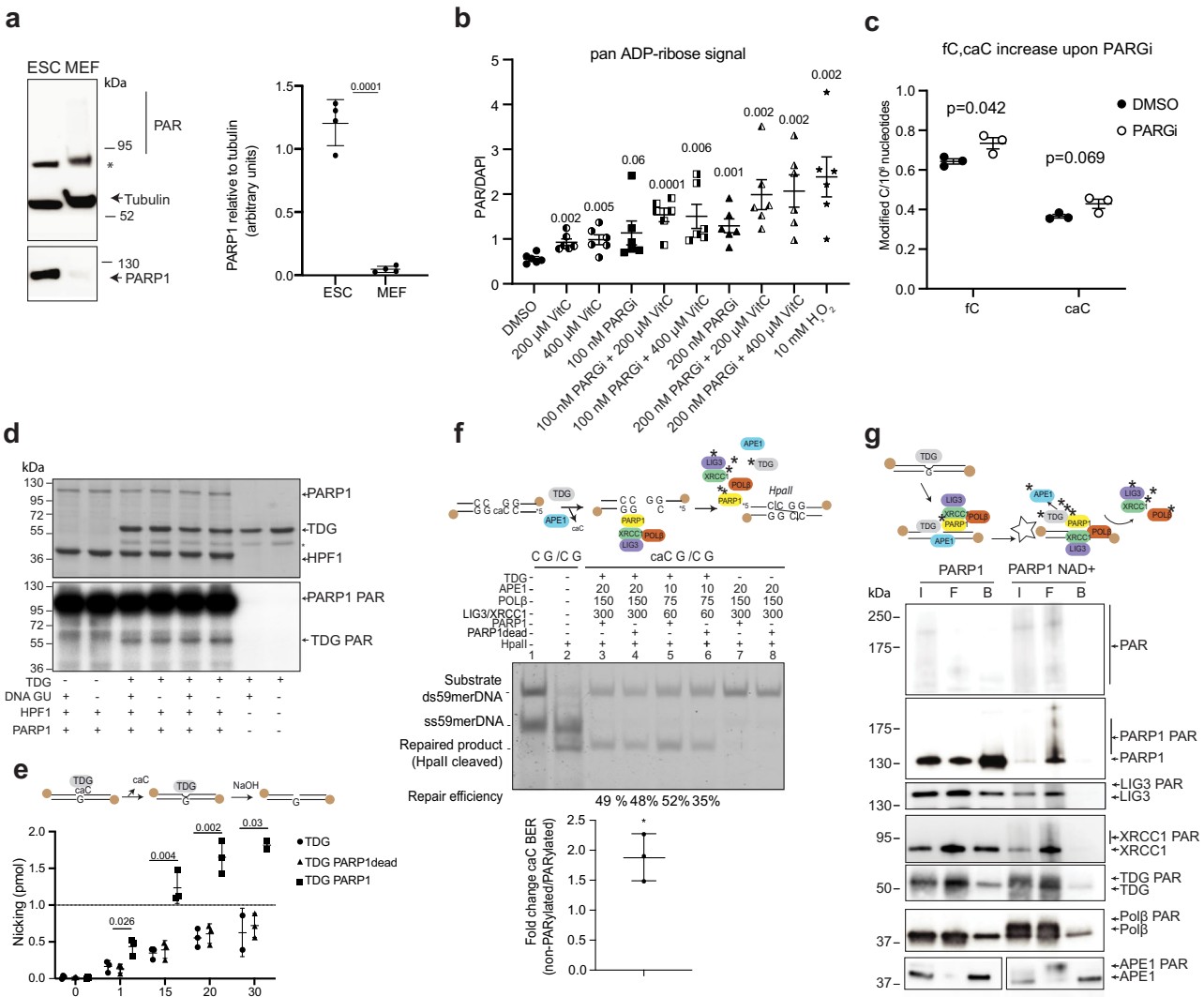

**Fig. 1 | Covalent PARylation of BER proteins stimulates DNA demethylation-associated caC-excision. a** Immunoblot of whole cell extracts of mESC and mouse embryonic fibroblasts (MEF) showing PARP1, poly(ADP-ribose) (PAR) and Tubulin as loading control. Quantitation of PARP1 relative to tubulin. (mean values ± SD, $n = 4$ biological replicates. T test, two tailed) * unidentified band. **b** pan-ADP-ribosylation signal (normalized to DAPI) of ESCs upon indicated stimuli for 16 h. $H_2O_2$ treatment was performed on ice for 10 min. (mean ± SEM, $n = 6$ images with each 10-20 mESC). **c** Mass spectrometry measurements of fC and caC in mESC treated with DMSO or 25 nM PARGi for 16 h. (mean ± SEM of $n = 3$ independent biological replicates). **b, c** Numbers above scatter plots indicate $p$-values of two-tailed t tests per condition compared to DMSO. **d** In vitro PARylation of TDG. Purified full-length human TDG was incubated with PARP1 and HPF1 in the presence of [32 P]-NAD$^+$, double-stranded DNA oligomer and G•U containing DNA oligonucleotides as indicated. Proteins were separated by SDS-PAGE and analysed by Coomassie blue staining (upper panel) and autoradiography (lower panel). Asterisk indicates truncated TDG protein. **e** Effect of TDG PARylation on catalysis in a base release assay. Full-length TDG was incubated with G•caC mismatched DNA, the AP-sites generated cleaved by the addition of NaOH at indicated times and the

products analysed by denaturing gel electrophoresis and fluorescence detection. Dotted line, turnover threshold where 1 pmol substrate per 1 pmol TDG is processed. (mean ± SD, $n = 3$ independent experiments, TDG; TDG PARP1 at timepoint 30 min $n = 2$ independent experiments); two tailed t test, numbers above data points indicate $p$-values. **f** Reconstitution of the BER side of TET-induced active DNA demethylation. 59mer double-stranded DNA substrates with methylation-sensitive *HpaII* restriction site (lane 1), digested by *HpaII* (lane 2), *HpaII* methylation-sensitive restriction site with a caC (caCG/CG), not cleavable by *HpaII* (lane 7–8), repaired product (CC/CG) cleavable by *HpaII* (lane 3-6). Percentage of repaired product is indicated. ss59merDNA, single-stranded DNA. Numbers above graph indicate protein concentration in fmol, for POLβ in pmol. Quantitation of relative caC BER, $n = 3$ independent experiments. Fold change relative caC BER compared to no change =1, mean ± SD, two tailed t test, *$p = 0.017$). **g** TDG-BER PARylation analysis. Purified TDG, LIG3, XRCC1, POLβ, and APE1 proteins bound to uracil-containing DNA substrate immobilized on streptavidin (star) were incubated with PARP1 and HPF1 as indicated. I input, F flow, B bound; $n = 3$ independent experiments. Source data are provided as a Source Data file.

but also the covalent PARylation of TDG and all proteins of the BER machinery. PARylation reduces DNA affinity of the BER proteins and PARP1, facilitating their dissociation from DNA after completion of the repair process. This allows the repair proteins to re-engage in a new cycle of BER, thus increasing the overall DNA demethylation efficiency. We conclude that covalent PARylation modulates the dynamics of protein-DNA interactions in BER to promote TDG dependent active DNA demethylation in ESCs.

## Results

### BER mediated active DNA demethylation in mESC depends on dynamic PARylation

PARP1 was reported to be highly expressed in mESC and required for early development[12,13]. We confirmed by immunoblotting very high levels of PARP1 protein in extracts of pluripotent mESC when compared to mouse embryonic fibroblasts (MEFs) (Fig. 1a). PAR synthesis was low in unchallenged cells but became highly activated after

induction of oxidative DNA damage by $H_2O_2$ as reported before[11,14–16]. This PARylation response was absent in mESC treated with the PARP1 inhibitor Talazoparib[17] and, hence, dependent on active PARP1 (Supplementary Fig. 1a). We therefore conclude that high PARP1 protein levels are characteristic for pluripotent mESC, and that PARP1 in mESC is activated in response to DNA base damage.

To test whether TET-TDG dependent active DNA demethylation is associated with PARP activation, we treated mESC with Vitamin C (VitC, 200 μM, 16 h)[18,19]. In mESC, VitC increased the levels of Tet-mediated oxidation of hmC by ~2.5 fold, fC by ~8.5 fold and caC by 11.5 fold when compared to untreated controls (Supplementary Fig. 1b). Notably, VitC also increased levels of ADP-ribose in mESC in a concentration dependent manner (Fig. 1b, Supplementary Fig. 1c). This shows that VitC stimulated active DNA demethylation is associated with PARP activation in mESC. PARylation is a reversible protein modification dynamically regulated by PARP and PAR glycohydrolase (PARG) activities[20]. Treatment of mESC with PARG inhibitor (PARGi; PDD00017273; 200 nM) increased PAR levels, as expected (Fig. 1b). This correlated with a moderate but reproducible increase of global fC and caC levels (Fig. 1c), indicating that de-PARylation by PARG activity is required for efficient turnover of fC and caC by TDG-BER. Notably, the increase of fC and caC cannot be explained by the reported inhibitory effect of PAR/PARP1 on TET activity, according to which PARG inhibition would be expected to reduce rather than increase TET activity in cells[21,22]. Moreover, treatment of mESC with both VitC and PARGi further increased PAR levels to varying degrees when compared with single treatments (Fig. 1b), whereas fC and caC levels remained largely unchanged (Supplementary Fig. 1d). Together, these results show that active DNA demethylation in mESC induces PARP activity and that de-PARylation by PARG is important for sustained BER of fC and caC.

## Covalent PARylation of BER proteins stimulates active DNA demethylation

BER in the context of active DNA demethylation starts with the excision of oxidized mC bases by TDG. Human TDG was reported to be PARylated at Ser85 in its N-terminal domain[12] but the function of this modification was never addressed. The TDG N-terminus cooperates with the catalytic CORE domain in DNA substrate and AP-site binding, and thereby modulates substrate recognition and enzymatic turnover of the glycosylase[23,24]. To investigate the functional impact of BER protein PARylation, we first validated the ability of PARP1 to covalently modify TDG using an established in vitro assay[25,26] (Fig. 1d, Supplementary Fig. 1e). TDG was PARylated in this assay and PAR was attached to both its N-terminal and CORE domains (Supplementary Fig. 1e). Next, we addressed the function of TDG PARylation in a caC base excision assay. TDG binds with high affinity to the AP-site product following excision of a DNA base, which is the rate limiting step in TDG-initiated BER[27,28]. As expected, full-length TDG excised a caC opposite guanine from an oligonucleotide DNA duplex but the reaction was inhibited at a product/enzyme ratio lower than 1[24,29] (Fig. 1e). PARylation by active PARP1 stimulated caC excision activity of TDG in comparison to reactions with a catalytically deficient PARP1dead, which binds DNA[30], or no PARP1 (Fig. 1e, Supplementary Fig. 1f). The reaction with active PARP1 achieved product/enzyme ratios significantly higher than 1, suggesting that PARylation of TDG enhanced its rate-limiting AP-site dissociation step (Fig. 1e).

To assess the effect of covalent PARylation on TDG-BER mediated active DNA demethylation, we reconstituted the complete process with purified recombinant TDG, APE1, POLβ, XRCC1 and LIG3 (Supplementary Fig. 1g), using a DNA oligonucleotide substrate containing a *HpaII* recognition site (CCGG) in a hemi-carboxymethylated configuration (CcaCGG, Fig. 1f)[29]. *HpaII* is CpG methylation sensitive and cannot cleave a CcaCGG sequence (Fig. 1f, lanes 7 and 8), but caC repair by TDG-BER results in the restoration of a cleavable CCGG site (Fig. 1f,

lanes 3-6). caC repair to unmodified Cs with recombinant TDG, APE1, POLβ, XRCC1 and LIG3 was significantly more efficient (>1.8 fold, SD ± 0.3) in the presence of active PARP1 and $NAD^+$ than in the presence of PARP1dead (Fig. 1f, lane 5 and 6; Supplementary Fig. 1h). These data show that PARylation promotes enzymatic turnover in TDG BER to accelerate caC repair.

The coordinated handover of DNA repair intermediates from one enzyme to another is an important feature of BER and has been conceptualized in a "passing the baton" model[31], where protein-protein and protein-DNA interactions are proposed to regulate the rate of individual repair steps. TDG dependent BER appears to be a special case, involving posttranslational SUMO modification to regulate AP-site dissociation[27], possibly due to the specific need for AP-site protection in the context of DNA demethylation[29]. To investigate the mechanism by which covalent protein PARylation stimulates TDG BER, we reconstituted the entire process with purified TDG, XRCC1, POLβ, APE1, LIG3 and PARP1, a double biotinylated G•U containing DNA duplex immobilized on streptavidin beads and dCTP to allow for DNA repair synthesis. In this setup, we measured the binding of the BER components to the DNA substrate in the presence or absence of PARylation (±$NAD^+$). Input (I), unbound (F) and bound proteins (B) were analysed by immunoblotting. Auto-PARylation of PARP1 was readily observed in the presence but not in the absence of $NAD^+$ (Fig. 1g). In the presence of $NAD^+$, TDG enrichment on the DNA was reduced compared to the reactions without $NAD^+$ (Fig. 1g). The fraction of TDG that remained DNA-bound seemed unmodified, while the unbound fraction was mainly PARylated. Notably, all other BER factors (APE1, XRCC1, LIG3, POLβ) in the assay as well as PARP1 itself behaved the same as TDG upon PARylation; DNA substrate binding was significantly reduced and the bound fraction consisted of primarily unmodified proteins (Fig. 1g). These findings show that PARylation reduces the affinity of TDG and downstream BER factors to the DNA substrate under repair, stimulating their dissociation and turnover and, thus, promoting the overall process of DNA demethylation.

## BER generated AP-sites and SSBs activate PARP1 to PARylate and dissociate BER proteins from DNA

The role of PARP in sensing AP-site and SSB intermediates of BER is strongly implicated but not understood in mechanistic detail[32]. Biochemical reconstitution of TDG-initiated BER (Fig. 1f) showed a clear engagement of PARP1 activity but it remained open whether this concerns the TDG dependent base excision step specifically or applies equally to the downstream AP-site or SSB repair steps. To address this, we prepared a defined AP-site by digesting a 60-mer G•U containing DNA substrate with *E.coli* uracil DNA glycosylase (UDG) for further processing by BER proteins XRCC1, POLβ, APE1, and LIG3 individually or as a pre-assembled complex (XRCC1-LIG3) in the presence of PARP1 and $NAD^+$. Combination of these components under BER assay condition induced variable PAR modification of all BER proteins, detectable as migration shifts in denaturing SDS-polyacrylamide gels (Fig. 2a). The higher molecular weight and more dispersed shifts observed with XRCC1 and LIG3 indicated strong PARylation of both proteins, the smaller and more discrete shifts of POLβ and APE1 could indicate either PARylation or mono-ADP-ribosylation (MARylation). None of the BER proteins was detectably modified in the presence of a PARP inhibitor (PJ34)[33] or a catalytic inactive PARP1. Notably, PARylation was also not detectable when an intact, biotinylated homoduplex DNA was added instead of an AP-site containing substrate (Fig. 2a). This demonstrates that covalent PARP1-mediated PARylation of BER proteins is triggered by the presence of AP-sites.

Next, we assessed PARylation of BER proteins in a reconstituted AP-site repair assay[34] with purified APE1, POLβ, XRCC1, and LIG3 (Fig. 2b). PARP1 strongly PARylated XRCC1, POLβ, and APE1, and to a lesser extent LIG3. In the presence of homoduplex DNA, PARylation of XRCC1 was reduced and absent for APE1, POLβ, and LIG3. No

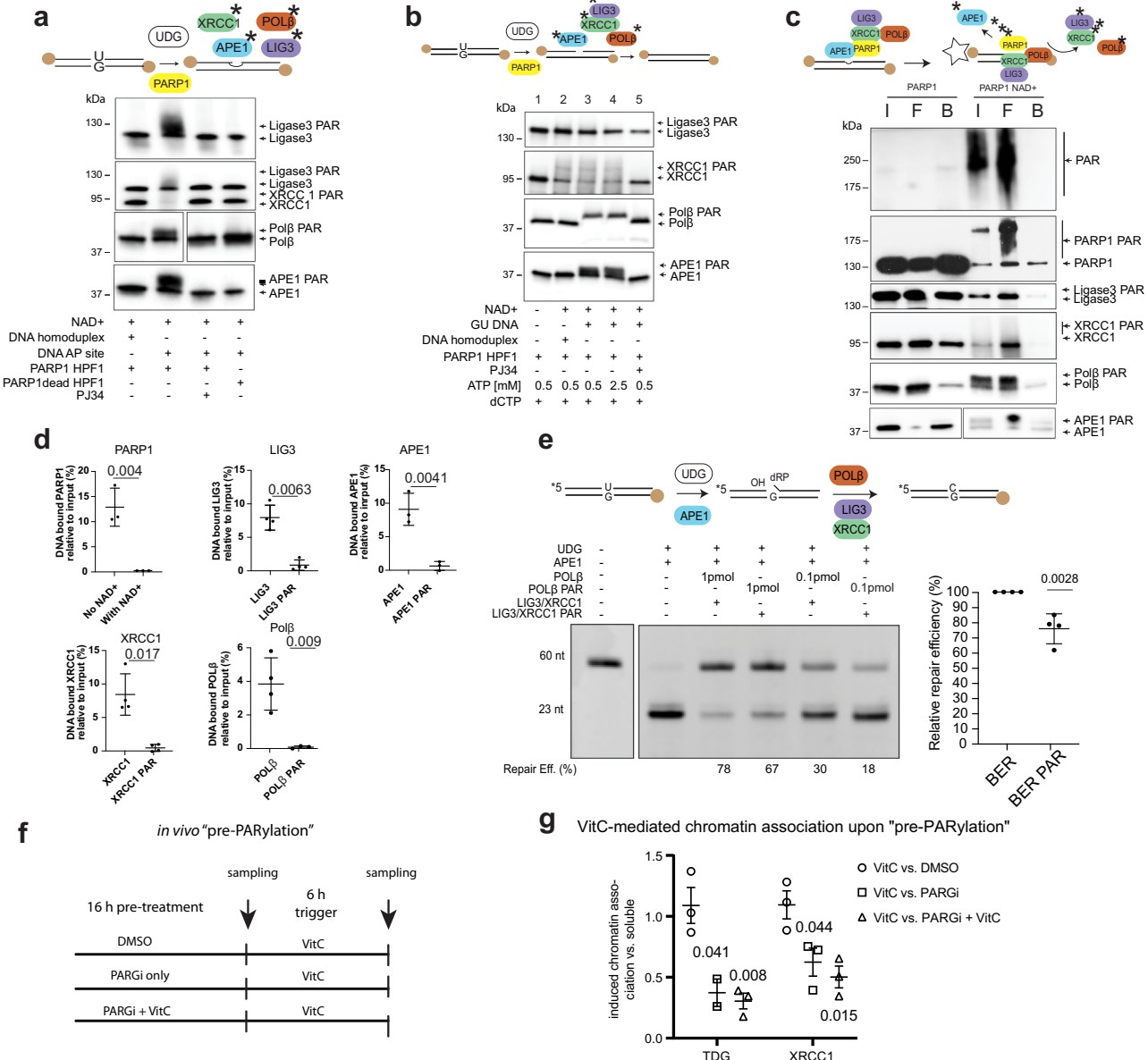

**Fig. 2 | AP-site and SSB activate PARP1 to PARylate BER proteins that then dissociate from DNA. a** In vitro PARylation of BER proteins; homoduplex or AP-site containing DNA substrate was incubated with either purified LIG3, XRCC1-LIG3 complex, POLβ or APE1 and purified active (PARP1wt) or catalytic inactive PARP1 (PARP1dead) and HPF1 as indicated. BER proteins were analysed by SDS-PAGE and immunoblotting. *n* = 3 independent experiments. **b** In vitro PARylation of BER proteins with PARP1 in reconstituted AP-site repair assay. Biotinylated uracil-containing DNA substrate was incubated with bacterial UDG to generate an AP-site for repair by APE1, POLβ, XRCC1 and LIG3 and equimolar amounts of purified active PARP1wt and HPF1. dCTP, ATP, and PJ34 PARP1 inhibitor were added as indicated. *n* = 3 independent experiments. **c** AP-site processing efficiency. BER proteins were incubated with DNA immobilized on streptavidin (star) beads and input (I), flow (F), and bound (B) proteins were analysed by SDS-PAGE and immunoblotting, *n* = 3 independent experiments. **d** Quantitation of DNA bound BER proteins (mean ± SD; PARP1 APE1, POLβ *n* = 3 independent experiments; LIG3 and XRCC1 *n* = 4).

**e** Reconstitution of the AP-site repair. APE1 cleavage of UDG-generated AP-site generates SSBs with a 3′-hydroxyl end and a 5′-end carrying the dRP residue, which is ultimately repaired by POLβ, XRCC1 and LIG3. Repair was performed using BER proteins or PARylated BER proteins as indicated. Percentage of fully repaired products is indicated. Quantitation of relative repair efficiencies of BER proteins and PARylated BER proteins (un-PARylated BER activity defined as 100%) (mean ± SD, *n* = 3 independent experiments). **f** Scheme for in vivo "pre-PARylation": VitC stimulates TET-mediated oxidation to engage BER/SSBR. PARGi inhibits removal of PAR from SSBR- (and DNA-) associated proteins. Combined pre-treatment aims to maximally PARylate the pool of available proteins. **g** Quantification of chromatin association (vs. soluble protein), comparing VitC vs. pre-treatment. (mean ± SEM, *n* = 3 independent experiments). Numbers in scatterplots represent *p*-values of unpaired, two-tailed *t*-test between the indicated and control condition. Source data are provided as a Source Data file.

PARylation of BER proteins was observed in the presence of the PARP inhibitor PJ34 or the absence of any DNA (Fig. 2b). These results demonstrated that PARP1 engages with general BER by sensing AP-site and/or SSB BER intermediates, which then activates covalent PARylation of XRCC1, APE1, POLβ, and LIG3.

Next, we addressed whether PARylation affects molecular interactions in context of AP-site repair as it did in the context of TDG-initiated BER (Fig. 1f). XRCC1 is well-known to physically interact with APE1, POLβ and LIG3 to coordinate BER[35,36]. We first tested whether PARylation affects the binding of XRCC1-POLβ to a DNA substrate[37].

We incubated a double biotin end-labelled G•AP-site DNA duplex[27] immobilized on streptavidin beads with XRCC1, POLβ, and PARP1 or PARP1dead and NAD⁺ (150 μM). Input (I), unbound (F) and bound proteins (B) were separated by SDS-PAGE and analysed by immuno-blotting. PAR synthesis was observed with PARP1 but not with catalytic inactive PARP1dead (Supplementary Fig. 2a). Again, the major fraction of XRCC1 was PARylated by PARP1 and failed to bind to the AP-site DNA (Supplementary Fig. 2a). In the presence of PARP1dead, XRCC1 was not PARylated and remained bound to the AP-site DNA. Likewise, PARy-lated POLβ failed to bind DNA, whereas unmodified POLβ, in presence of PARP1dead, bound to the AP-site DNA (Supplementary Fig. 2a). We next assessed the effect of PARylation on the concerted binding of XRCC1, POLβ, APE1 and LIG3 to an SSB-containing DNA substrate (Fig. 2c). In the presence of PARP1 but without available NAD⁺ and under physiological salt concentrations (150 mM NaCl) APE1, XRCC1, POLβ, and LIG3 were abundant in the SSB DNA-bound fraction (Fig. 2c). The same experiment in the presence of NAD⁺ resulted in PARylation and release of PARylated XRCC1, PARylated LIG3, PARylated APE1, and PARylated POLβ from the SSB substrate (Fig. 2c, d). Only under very low salt concentrations (20 mM NaCl) could the PARylated proteins be detected in the streptavidin DNA precipitate (Supplementary Fig. 2b). This demonstrated that the covalent attachment of PAR chains to DNA-engaged BER proteins reduces their DNA affinity and caused their dissociation from the DNA at physiological salt concentrations.

We then assessed whether PARylation alters the enzymatic activities of the core BER proteins, making use of specific biochemical assays based on a fluorescence-labelled substrate containing an AP-site generated by digestion of a 60-mer G•U-containing DNA duplex with UDG. PARylated APE1 (50% PARylated) and unmodified APE1, both fully processed the AP-site substrate available (Supplementary Fig. 2c, left). Likewise, PARylation of POLβ had no effect on its rate limiting AP-lyase activity when measured by AP-site cleavage (Supplementary Fig. 2d). We then reconstituted the entire BER process from AP-site incision to ligation. In this setting, APE1 and PARylated APE1 both efficiently cleaved the AP-site and POLβ incorporated 1 nucleotide (Supplementary Fig. 2d, e) irrespective of its PARylation state. Notably, addition of POLβ in slight excess (1 pmol) was required to yield high levels (>75%) of ligatable BER products (Supplementary Fig. 2e). Although PARylated POLβ (~50% PARylated, 0.1 pmol and 1 pmol) efficiently incorporated 1 nucleotide, ligation efficiency was slightly reduced compared to the reaction with non-PARylated POLβ (Supplementary Fig. 2c, right), suggesting a small effect of PARylation on the ligation efficiency by LIG3 (Supplementary Fig. 2e). Finally, we fully reconstituted BER with PARylated XRCC1(~50% PARylated), POLβ (~50% PARylated), LIG3 (~10% PARylated). Repair efficiency with PARylated proteins was significantly reduced (difference of means −26 ± 4.6%) when compared to assays with unmodified XRCC1/POLβ/LIG3 (Fig. 2e, Supplementary Fig. 2e). Altogether, this showed that the reduced DNA affinity of PARylated APE1, POLβ, and XRCC1 alters the overall repair kinetics of the general BER steps rather than the catalytic activities of the individual repair proteins, suggesting a contribution of PARylation to the active turnover of the BER complex.

## PARylation reduces the chromatin association of the BER proteins TDG and XRCC1

To test whether the reduced DNA affinity of PARylated BER proteins can also be observed in cells, we examined the effect of PARylation on the chromatin association of TDG and XRCC1. To this end, we fractionated nuclei of mESC and compared the ratio of chromatin bound vs. nuclear soluble protein following VitC induced mC oxidation by TET. The relative amounts of chromatin associated TDG and XRCC1 did marginally increase in mESC treated with VitC when compared to untreated cells (Fig. 2f, g, left, Supplementary Fig. 2f, top). Pre-treatment with PARGi or VitC plus PARGi to engage and PARylate BER proteins resulted in significant reductions of chromatin bound TDG

and XRCC1 (Fig. 2g, Supplementary Fig. 2f). This shows that inhibition of de-PARylation in mESC results in reduced chromatin association of TDG and XRCC1, which is in line with the biochemical evidence showing reduced DNA affinity of both PARylated proteins. These results strongly suggest that DNA and chromatin association of BER proteins is regulated by dynamic PARylation and de-PARylation.

## ADP-ribosylation of BER proteins interferes structurally with DNA binding

To understand how covalent PARylation of BER proteins modulates their interaction with DNA, we grafted MAR and PAR chains to the mapped acceptor residues of human BER proteins for which structural information for DNA interactions is available (APE1, POLβ, LIG3)[11], using the Rosetta Comparative modelling approach. Possible effects on overall protein conformation induced by MARylation/PARylation were not considered in the modelling. For APE1, we used the available APE1-AP-site DNA co-crystal structure[38]. Since this structure was derived from an N-terminally truncated APE1 that misses the MAR/PAR acceptor Ser26, we first modelled the full-length APE1, yielding a fold matching that of the truncated protein (>95%). In silico grafting of PAR chains (or MAR) to Ser26 in the flexible N-terminal redox domain generated a protrusion that sterically interferes with the interaction of APE1 with DNA (Fig. 3a). PAR chains (or MAR) modelled onto Cys138[11] protruded from the globular APE1 endonuclease domain (Fig. 3a, Supplementary Fig. 3a, b) where they unlikely interfere with DNA binding or catalysis (Supplementary Fig. 2d). POLβ consists of an N-terminal domain (8-kDA) that mediates DNA contact and removes the 5′- deoxyribose-phosphate (dRP) residue to trim the SSB 5′end for DNA ligation, and a C-terminal (31 kDa) DNA polymerase domain[39] (Fig. 3b). Modelling MAR and PAR to the POLβ-DNA co-crystal structure[40] revealed that ADP-ribosyl residues attached to Ser30 and/or Ser44[11] protrude from the N-terminal lyase domain in a way that interferes with the function of the helix−hairpin−helix DNA binding motifs of POLβ (Fig. 3b, Supplementary Fig. 3c)[39]. The model also predicts that PARylation of POLβ does not affect its interaction with the N-terminal domain of XRCC1 (Supplementary Fig. 3d)[41]. Grafting PAR to the mapped acceptor residues[11] near the DNA binding domain (DBD, His522) of the LIG3-DNA co-crystal shows that PARylation of LIG3 is likely to interfere with DNA binding as well[42] (Supplementary Fig. 3e). Structural modelling, therefore, suggests that covalent PARylation of BER proteins (APE1, POLβ, XRCC1 and LIG3) causes disruptive steric clashes with bound DNA, consistent with a negative impact on DNA binding affinity, favouring the dissociation of the BER complex from the repaired DNA.

## PARylation of XRCC1 reduces DNA damage association in cells

To validate the role of covalent XRCC1 PARylation in modulating protein-chromatin interactions in live cells, we generated U2OS cell lines expressing, besides endogenous hXRCC1, an ectopically encoded wild-type (hXRCC1wt-GFP) or PARylation-deficient human XRCC1 (hXRCC1pd-GFP). To generate hXRCC1pd-GFP, seven published PARylation acceptor serine residues (S103,184,193,219,220,236,268) and one arginine (R186) were each mutated to alanine (Fig. 3c); two more PARylation residues (S234, S259) were also reported to be phosphorylation sites and were therefore not mutated[11]. Recombinant hXRCC1pd was extracted and purified from 3 different clones and subjected to in vitro PARylation, revealing PARylation levels that were 50−70% lower when compared to hXRCC1wt (Supplementary Fig. 3f). Mutation of serine S268 within the reported nuclear location signal (NLS) did not change the nuclear presence of hXRCC1pd-GFP compared to hXRCC1wt-GFP (Supplementary Fig. 3g). Protein expression levels of the ectopic hXRCC1 variants amounted to ~70% of endogenous hXRCC1wt in two polyclonal cell populations (Supplementary Fig. 3h), and variance between individual cells was taken into consideration in the following experiment (Supplementary Fig. 3i).

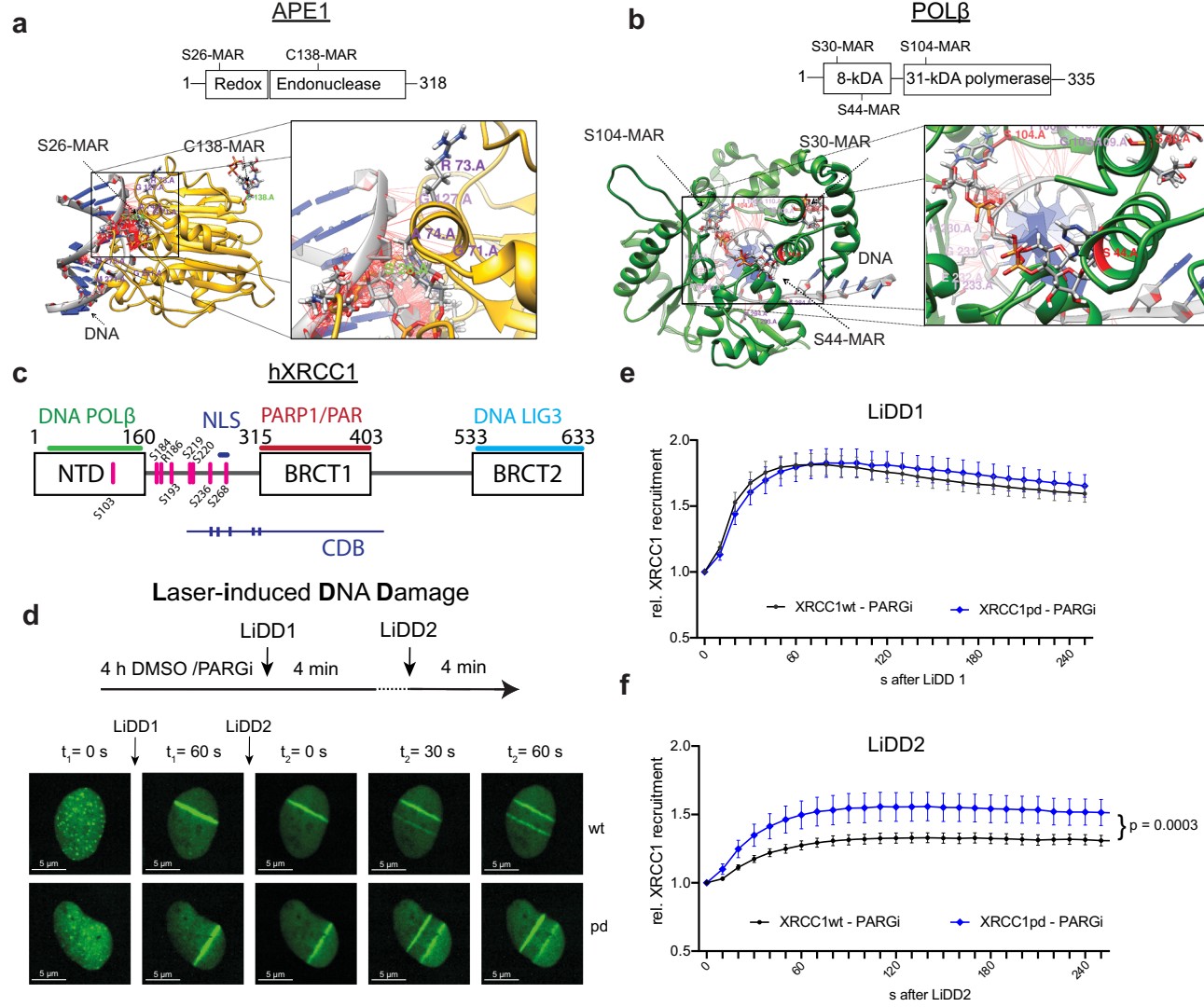

**Fig. 3 | Modelled covalent M/PARylation interferes with DNA binding and PARylation of hXRCC1 reduces recruitment to damage in U2OS cells.**
**a** Structure of human APE1 (PDB: 1DE8), with redox and endonuclease domains indicated. Modelled full-length APE1 structure with mono-ADP-ribose (MAR) at the acceptor residues S26 and C138[11]. **b** Structure of human POLβ (PDB: 6BTE), with 8-kDa and polymerase domains indicated. Modelled full length POLβ structure with MAR molecules at the acceptor residues S30, S44, S104[11]. Protein modelling with attached MAR structures was performed using Rosetta Comparative Modelling. **c** Schematic overview of hXRCC1 secondary structure. Structured domains are indicated in boxes; NTD: N-terminal domain, BRCT1/2: (breast cancer susceptibility protein). Relevant interactions with BER proteins and their localization are

indicated above the scheme. The position of reported PARylation sites that were altered for this study (hXRCC1pd) are indicated in pink. NLS: nuclear localization signal, CDB: central DNA binding domain with tick marks representing the crucial amino acid groups **d** Scheme of experimental procedure and representative images of U2OS cells treated with 200 nM PDD for 4 h, exposed to laser-induced DNA damage (LiDD) **e** Recruitment to site of first damage (LiDD1) in U2OS cells treated with 200 nM PDD for 4 h. **f** Re-recruitment to site of second damage (LiDD2). **e**, **f** Displayed are means, error bars indicate 95% confidence-interval based on $n = 31$(wt) and $n = 34$(pd) cells from 4 independent experiments. Source data are provided as a Source Data file.

Microlaser-induced DNA damage (LiDD) was shown to attract and engage DNA repair protein factors in a PARP dependent manner[43]. To investigate the dynamic properties of PARylation proficient and deficient GFP-tagged XRCC1, we performed LiDD recruitment assays in U2OS cells using a 355 nm (UV-A) laser as a damage source (Fig. 3d). The assay was designed to first engage a major fraction of cellular XRCC1 in repair and trigger PARylation by a first damage induction (LiDD1). Then a second damage (LiDD2) was induced and the recruitment of XRCC1 to this damage was monitored. XRCC1wt-GFP and XRCC1pd-GFP associated with equal dynamics to the sites of first DNA damage (Supplementary Fig. 3j), showing that the mutations introduced in XRCC1pd do not affect its DNA damage recruitment. Upon second irradiation (LiDD2) ~6 min after the first, XRCC1pd-GFP showed a slightly faster damage reassociation than XRCC1wt-GFP.

This suggested that engagement of XRCC1 with DNA damage alters its recruitment dynamics, presumably by covalent PARylation (Supplementary Fig. 3k).

To investigate the role of PARylation further, we performed the same experiment in the presence of a PARG-inhibitor (PARGi, PDD00017273 200 nM, 4 h) to stabilize protein PARylation. Again, hXRCC1wt-GFP and hXRCC1pd-GFP were recruited with the same kinetics after LiDD1 (Fig. 3e). After LiDD2, however, PARG inhibition delayed the damage association of XRCC1wt-GFP significantly more than that of the PARylation deficient XRCC1pd-GFP, which relocated with significantly faster (~25%) kinetics (Fig. 3f). These findings are in line with the biochemical results and show that a previously engaged and therefore pre-PARylated hXRCC1, is less likely to re-engage with new damage and requires de-PARylation to regain optimal damage affinity.

## PARylation promotes the repair of DNA demethylation- and DNA damage-associated SSBs

To investigate the stimulatory role of PARylation on DNA demethylation-associated and general BER in a cellular context, we measured genome-wide SSB formation in wildtype and TDG depleted (TDGnull) mESC[44], treated or not with the PARP1 inhibitor Talazoparib at a concentration (5 nM) inducing no detectable PARP1 trapping (Supplementary Fig. 4a). Following nick-translation with digoxigenin-modified nucleotides, immunoprecipitation and high-throughput sequencing (SSB-seq, n = 3)[45], we identified >28,000 genomic regions in wildtype mESC showing a significant enrichment of SSBs relative to input (log2FC ≥ 2, p ≤ 0.0001, Fig. 4a). In general, the genomic distribution of these SSBs showed a strong preference for gene regulatory and genic regions, including gene promoters (16% of SSB peaks), introns (28%) and exons (15%) compared to intergenic regions (12% of peaks). This is in line with previous observations in human cells[46,47] (Supplementary Fig. 4b). The high proportion of non-randomly positioned SSBs implicates the action of a SSB targeting mechanism that superimposes the stochastic generation of SSBs through random DNA base or backbone damage, or topoisomerase action[7,48,49]. To investigate the fraction of SBBs that may relate to TET-TDG mediated active DNA demethylation, we compared their occurrence with mapped genomic locations of TDG-dependent caC excision[50]. This identified 6843 regions where SSB enrichment (24% of SSB peaks) coincides with caC excision in mESC (Fig. 4a, c). Similarly, we used a genomic map of TET-mediated oxidation of CpGs (i.e., dynamic CpGs)[51] in mESC and observed that 10,583 SSB-enriched regions (37% of SSB peaks) coincide with CpGs displaying high (above-median) TET-activity (Fig. 4a). Hence, despite the differences in mESC background and culturing conditions underlying these analyses, 51% of detected SSB peaks in our cells can be associated with sites of active DNA demethylation.

Notably, following PARP1 inhibition by Talazoparib, SSBs were proportionally reduced at peaks of caC-excision (18% in Tal vs. 24% in DMSO) and shifted towards more distal (>−800 bp and +800) sites (Fig. 4 b, c). Likewise, PARP1 inhibition reduced the association of SSB-enriched regions with dynamically methylated CpGs (29% in Tal vs. 37% in DMSO) (Fig. 4d)[51]. Quantitatively, PARP1 inhibition reduced the SSB signal (i.e., normalized read counts) at sites of TDG-dependent caC excision to levels observed in induced TDGnull mESC (Fig. 4e). Moreover, TDGnull mESC did not show an effect of PARP1 inhibition on SBB occurrence at the same sites. The significant loss of SSB enrichment at sites of TET-TDG-BER mediated mC oxidation and repair upon PARP1-inhibition corroborates an engagement of PARP1 in active DNA demethylation and is consistent with PARylation promoting TDG turnover; inhibition of TDG turnover will reduce caC excision and, thus, AP-site and SSB formation.

To contrast the situation at genic sites undergoing active DNA demethylation with sites where spontaneous, TET-TDG independent DNA damage is expected to predominate, we analyzed a subset of 100,000 randomly chosen simple repeat regions (e.g., microsatellites). These repeat regions displayed a significantly lower basal level of SSB formation than sites of caC excision (Fig. 4e, f) and PARP1 inhibition induced a significant increase rather than a decrease of SSB signal (Fig. 4f). The same was true for regions with a GC-skew[52] outside of gene regulatory regions (Fig. 4g). GC-skewed regions in promoters (Fig. 4h), on the other hand, and enhancers (Fig. 4i) showing BER-mediated caC excision, or mC-oxidation activity (Supplementary Fig. 4c, d), however, again showed a decrease of SSBs upon PARPi treatment. Analyzing the SSB signal at enhancers without DNA demethylation activity, revealed also an increase of SSB generation in response to PARPi treatment (Supplementary Fig. 4e). The relative increase of SSB formation upon Tal at sites where little or no active DNA demethylation is detectable is consistent with PARylation promoting turnover of the general BER factors; inhibition of this turnover will lead to an accumulation of spontaneously generated AP-sites and SSBs.

Given the pronounced enrichment of SSBs at promoters, we examined the functional relationship of PARP1 inhibition, active DNA demethylation and regulation of gene expression. We measured expression levels of genes with promoters showing SSB enrichment and/or caC excision. Genes displaying caC-excision in their promoter are generally expressed at higher level than genes without detectable caC excision and this effect is increased for genes where both caC excision and SSB enrichment are detectable (Supplementary Fig. 4f), consistent with active DNA demethylation in promoters driving transcription[51].

Overall, these results demonstrate that PARylation by PARP1 promotes active DNA demethylation at gene regulatory regions in mESC, generating high levels of TDG-BER-associated, "programmed" SSBs that correlate with increased transcriptional activity. Outside regions undergoing active DNA demethylation, however, PARP1 activity engages in the repair of spontaneous DNA damage as evident from increased SBB accumulation at such sites upon PARP1 inhibition.

## Discussion

While it is known that PARP1 ADP-ribosylation accelerates AP-site or SSB repair[4], the potential role of PARP1 in sensing BER related SSBs is still under debate because DNA intermediates of BER have been proposed to be passed on from one enzymatic step to the next within the repair complex[31]. The data presented in this study, however, demonstrate that PARP1 can sense AP-site and single-strand breaks (SSBs) produced during BER in the context of active DNA demethylation. This is in line with previous work demonstrating that PARP1 is important for SSB sensing during BER of alkylation DNA damage[53].

It is well established that PARP1 and PARP2 are critical for the recruitment of the BER scaffold protein XRCC1 to SSBs to facilitate their repair[6]. However, it is still not known which proteins are PARylated and bound by XRCC1 in the context of BER. Notably, PARylation was shown to facilitate the interaction of XRCC1 with the SUMO ligase TOPORS which promotes XRCC1 SUMOylation and recruitment of POLβ[54]. This observation is in line with previous observations that SUMOylation of XRCC1 is important for TDG-BERosome assembly and coordination of TDG BER activity during DNA demethylation in differentiating mESC[2]. Our data now show that AP-sites produced by TDG during active DNA demethylation trigger PARP1 activity and it is a likely, although hypothetical, scenario that PARylation recruits TOPORS to SUMOylate XRCC1 and thus promote TDG-BERosome formation. The concept that a sequence of non-covalent PAR mediated interactions modulate SUMO mediated BERosome assembly is intriguing and warrants further investigation.

Our data also show that PARylation occurs further downstream in the BER/SSBR process, involving the covalent modification of all core BER proteins. This applies to both, BER of general DNA base damage and TDG-dependent active DNA demethylation and the data indicate a function in BERosome dissociation. Mechanistically, covalent PARylation reduces the affinity of BER proteins to DNA/chromatin and results in the dissociation of the BER/SSBR complex (Fig. 4j), as previously shown for single proteins including PARP1 itself[55] but also the nucleotide excision repair protein RAD23B[56]. Hence, for BER to be functional, covalent PARylation of BER factors (and PARP1) must be tightly regulated in space and time to avoid premature dissociation of the repair complex. In this regard, we reason that the dissociation effect by covalent PARylation is counterbalanced by the non-covalent interaction of BER proteins with PAR chains on chromatin and by the engagement of factors like PARG and XRCC1. Both PARG and XRCC1 are recruited quickly to DNA damage following PARP activation. Published evidence suggests that PARG may then antagonize[57] and XRCC1 limit PARylation activity (XRCC1)[58] to fine-tune the BER process.

The biological significance of PARP1-mediated PARylation in general BER and in active DNA demethylation in mESC is apparent in the SSB profiling data. SSBs do occur randomly but more than

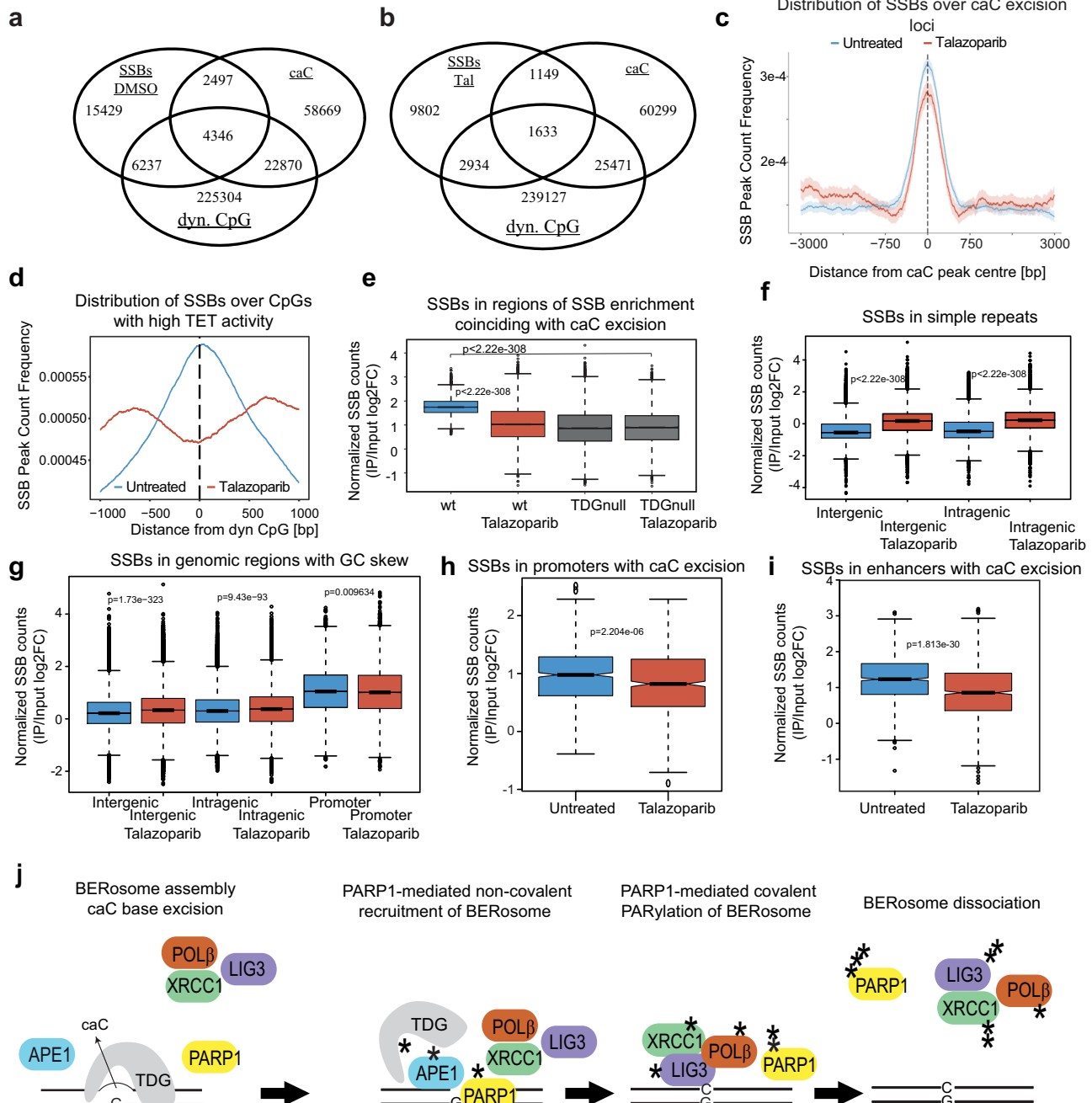

**Fig. 4 | PARylation promotes targeted BER in DNA demethylation and the repair of random SSBs in mESC. a** Overlaps of SSB-enriched regions in mESC with sites of active DNA demethylation. caC, sites of caC accumulation in TDGnull mESC[50]; dyn.CpG, CpGs undergoing above-median rates of TET-mediated oxidation[51]. **b** Same as **a** but with SSB-enriched regions from Talazoparib-treated mESC. **c** Mean frequency of SSB-enriched regions over caC excision loci. Lighter shadings indicate the 95% confidence interval **d** Mean Frequency of SSB-enriched regions over CpGs undergoing above-median TET-mediated oxidation. **e** Normalized SSB read counts in regions of SSB enrichment coinciding with caC excision, n = 6877[50]. **f,** Normalized SSB read counts in 100,000 randomly chosen simple repeats (repeat masker, UCSC). **g** Normalized SSB read counts in regions with increased GC-skew[52] (intergenic n = 131,407, intragenic n = 109,507, promoter

n = 22,898). **h** Normalized SSB read counts in promoters (TSS −2 kb + 500 bp) exhibiting caC excision and SSB enrichment in unchallenged mESC (n = 751). **i** Normalized SSB read counts in enhancers (Fantom) exhibiting caC excision and SSB enrichment in unchallenged ESCs (n = 10,972). All data originate from n = 3 independent SSB-seq samples. **j** Model of PARP1-driven active DNA demethylation. AP-sites and SBBs generated in TET-TDG initiated DNA demethylation are sensed by PARP1, which is then activated. PARP1 PARylates TDG and the core BER enzymes, promoting their dissociation from DNA and turnover. Boxplots represent the mean and center quartiles within the box. Whiskers range to the 25 and 75 percentiles, respectively and circles represent values outside of the box. Notches approximate 95% confidence interval. Source data are provided as a Source Data file.

expected show a high enrichment at specific genomic loci such as gene promoters and enhancers where TET proteins are highly active, DNA demethylation is ongoing and transcriptional activity is high. At these sites, PARP inhibition reduced detectable SSB levels. This is, at first sight, counterintuitive as PARP inhibition is expected to reduce the

recruitment of SSB repair proteins and, hence, increase levels of unrepaired SSBs. The observation, however, is consistent with our biochemical data on TDG-initiated BER, showing that PARylation accelerates TDG base release activity, and thereby SSB formation, during BER-mediated active DNA demethylation. By contrast, our SSB

profiling shows that PARP1 inhibition accumulates SSBs at genomic sites where no detectable active DNA demethylation is occurring, but different forms of genomic instability can be observed (e.g., simple repeats including microsatellites and GC-skewed sequences outside of gene regulatory regions). This is consistent with the current view that PARylation is necessary to recruit DNA damage response factors to sites of spontaneous DNA damage, and with our biochemical data showing that covalent PARylation is required to turnover engaged BER/SSBR.

Together, these observations confirm that non-covalent recruitment via PAR chains is a rate limiting step in the engagement of BER/SSBR proteins at sites of spontaneous DNA base damage or SSB formation, whereas covalent PARylation of BER and SSBR proteins can become limiting at sites where BER proteins are abundant, such as gene regulatory regions where dynamic DNA methylation continuously generates clustered BER substrates. We, therefore, propose that PARylation has a dual role in promoting enzymatic turnover in active DNA demethylation and in SSBR. Beyond its well-established role in facilitating the repair of spontaneous, "unscheduled" DNA breaks, PARylation is an integral part of targeted BER in the context of active DNA demethylation, where it regulates the coordinated action of TDG and the BER proteins.

Proteome analysis indicated that several DNA glycosylases (i.e., MUTY, MPG, MBD1/2/3/4, NEIL1/3, NTHL1, UNG) are also PARylated following exposure of cells to oxidative stress[11], suggesting that the mechanisms proposed here represent a general feature of BER. The concept of a PARylation controlled assembly and disassembly of protein complexes provides a mechanistic framework to explain the coordination and the dynamics of complex multi-enzyme transactions involved in DNA repair and the maintenance of genomic function and warrants further investigation.

## Methods

### Cell culture

mESC were grown on feeders at 37 °C for two passages in serum-containing mESC medium in a humidified atmosphere containing 5% $CO_2$. Then mESC were generally grown without feeders 2i medium containing LIF. For IF (Supplementary Fig. 1a), mESC were treated with DMSO or 100 nM Talazoparib for 12 h. Then mESC were treated with 1 mM $H_2O_2$/$MgCl_2$ for 10 minutes to induce PARylation and harvested by adding modified RIPA buffer[59]. 10 µM PJ34 and 75 µM tannic acid in modified RIPA buffer prevent PARP1/PARP2 mediated ADP-ribosylation and PARG mediated hydrolysis of poly-ADP chains respectively during the sample preparation until acetone precipitation.

For nuclear fractionations and oxmC quantification, mESC were pretreated with 25 nM PDD 00017273 (Medchem Express) ± 200 µM Vitamin C (L-ascorbic acid BioXtra, Sigma), for 16 hrs followed by another treatment with 200 µM Vitamin C for 6 h. Cells were harvested and dissociated with Accutase (ICT). U2OS cells were cultured in DMEM (high glucose, Sigma) containing 10% FBS (FBS supreme, PAN Biotech) and 1× GlutaMAX (Thermo Fisher scientific).

### Immunostaining and microscopy

1a: Cells grown on glass cover slips were treated with or without 50 nM Tal for 30 min followed by 10 min treatment with 0.5 mM $H_2O_2$ and fixed with methanol/acetic acid (3:1) for 5 min on ice. After fixation, cells were blocked by 5% milk and 0.05% Tween 20 in PBS. After blocking, cells were first stained with anti-PAR Ab (10H) followed by Cy3-conjugated goat anti-mouse secondary Ab (Jackson ImmunoResearch). Finally, cells were stained with Hoechst 33258 stain solution and mounted with Vectashield mounting solution. Fig. 1b, Supplementary Fig. 1c: ESCs were grown in IBIDI chambered coverslips. PAR-staining with the pan-ADP-ribose detection reagent (MABE1016, Merck) was performed according to reference[60] using a dilution of 1:200.

Images were acquired using a Leica SP5 scanning confocal microscope equipped with a 63×/1.4 HCX PL Apo CS objective, utilizing a scanning rate of 400 Hz and an averaging over 2 frames at a resolution of 512 × 512 pixel. Pan-ADP-ribose signal intensity was measured over the whole image of z-projections (summed intensity) using FIJI(1.53t), and normalized to the corresponding DAPI signal.

### Live cell imaging and damage-recruitment experiments

Recruitment of GFP-tagged XRCC1 to sites of UV-A laser-induced DNA damage (LiDD) was assessed by spinning disc (Yokogawa CSU-W1-T2, 50 µm pinhole) confocal microscopy under controlled temperature (37 °C) and $CO_2$ (5%) atmosphere (okolab). The system consists of a fully automated Nikon Ti-E microscope, equipped with a Hamamtsu Flash 4.0 V2 CMOS camera and a passively Q-switched 355 nm UV-A ablation laser (2D-VisiFRAP-DC, Visitron Systems GmbH) with 10 Hz to 2 kHz repetition rate, pulse width of 400 ps and 1 µJ/pulse at 4 kW peak power. Polyclonal U2OS cells, stably expressing hXRCC1wt or hXRCC1pd C-terminally fused to GFP, were seeded at densities of $15-30 \times 10^3$ cells/cm² into 18-well ibidi µ-slides (ibidi GmbH) and treated with 200 nM of PARGi (PDD-0017273 MedChemExpress) 4 h before assessment. Moderate levels of UV-A-induced DNA damage were induced along a transversal line in the nuclei of 8-10 randomly chosen cells per image section by applying 2.5 and 0.5 ms pulses for LiDD1 and LiDD2, respectively, using a ND4 gray filter. LiDD1 was induced 6 min before LiDD2, while 2 time lapse confocal imaging series with 10 s intervals were recorded for 4 and 10 min, respectively, using a 60× water immersion objective (Nikon CFI Plan Apo VC 60XC WI) with a 488 nm laser line, a 525/50 GFP emission filter, 1000 ms exposure time and 4 × 4 camera binning.

Image stacks of each time lapse series (512 × 512 px, 312 px/cm, 16-bit gray scale) were analyzed with the Fiji image processing package (https://fiji.sc/)[61]. After background subtraction (settings: <20 px rolling ball radius > , <sliding parabolic > , <disable smoothing > , nuclear ROIs were determined by automatic thresholding (setting: <default > ) of z-projections (setting: <Standard deviation > ), while ROIs for LiDD1/2 were manually defined, covering approximately 15-20% of the nuclear ROIs. The plugin "Multi Measure" (https://www.optinav.info/Multi-Measure.htm) was used to measure the integrated density of gray values for each ROI and stacked image/timepoint. To account for bleaching, the ratios of the integrated densities of LiDD and nuclear ROIs were calculated for each timepoint and normalized to pre-LiDD images. The XRCC1-GFP recruitment data to LiDDs of >30 nuclei obtained from 4 independent experiments was statistically analyzed in GraphPad Prism (v. 9.5.1). Considering $p$-values < 0.05 as significant, the dynamics of recruitment was analyzed by the RM two-way ANOVA model with genotype of XRCC1 and the time as parameters, while the values of each timepoint were compared post-hoc by the Šídák's multiple comparisons test.

### Bioinformatic modeling of BER protein structures

Rosetta-commons comparative modeling algorithm with multi-templates approach was used to generate full protein structure, for the truncated crystal structure of human BER proteins. The crystal structures used in the modeling approach are for XRCC1 (3k77, 3qvg, 3k75), LIG3 (3l2p, 6wh1, 3pc7, 3l2p), polB (3lqc, 1bpx, 4nln) and APE1 (1de8, 1e9n, 6w2p). Pymol and Chimera were used to add mono-ADP-ribose to the reported acceptor sites and structural alignment again with the selected crystal structures for visualization purposes and image creation. Mono- and poly-ADP-ribose chains were grafted to the in vivo mapped ADP-ribose acceptor sites of human BER proteins[11].

### Protein interaction and "Far-Western" analysis

"Far-Western" analysis was performed by spotting purified human POLβ (0.5 µg), APE1 (0.5 µg), LIGIII (0.2 µg), XRCC1 (0.2 µg), TDG full size or TDG domains, TDG SUMO (0.5 µg) and BSA (1 µg) on

nitrocellulose membrane (Protran, Amersham). PARP1 was incubated prior to probing for 5 min at 37 °C in PARylation buffer (50 mM Tris pH 8.0, 0.5 mg/ml BSA, 1 mM DTT, MgCl$_2$ 2 mM) in the presence or absence of NAD$^+$ (1 mM), with homoduplex DNA (25 nM). Membranes were then incubated with purified PARP1 or PARylated PARP1 (1 μM) in incubation buffer (50 mM Tris pH 8.0, 10% glycerol, 150 mM NaCl, 0.5 mg/ml BSA, 0.1% NP-40, 1 mM DTT) at 4 °C for 4 h. Unbound PARP1 protein was removed by brief washing and membranes were probed with polyclonal anti-PARP1 antibody (ab32138 Abcam (rabbit) 1:1000, dilution) in non-fat dry milk TBS (100 mM Tris−HCl pH 8, 150 mM NaCl) 0.1% Tween-20 (Sigma) and analysed by chemiluminescence detection using PXi imaging system, Syngene.

## PARylation reaction

BER proteins (0.2 μg) were PARylated separately or as a BER complex (0.2 μg each protein) on DNA (10 pmol) with PARP1wt or PARP1dead (0.1 μg) and HPF1 (0.1 μg), for PARylation specificity[25,62], as indicated. PARylation reaction was perfomed in PARylation buffer (50 mM Tris pH 8.0, 0.5 mg/ml BSA, 1 mM DTT, MgCl$_2$ 2 mM) in presence or absence of NAD$^+$ (150 μM−1 mM), for 15 min at 37 °C. AP-site substrate was generated by digestion of uracil containing homoduplex oligonucleotides with uracil DNA glycosylase (UDG, 5 units New England BioLabs) in PARylation buffer, 5 min at 37 °C. The reaction volume was 20 μl. To generate the SSB substrate, the AP substrate was incubated with APE1 in PARylation buffer, 5 min at 37 °C. In vitro PARylation of TDG with radioactive labelled NAD$^+$. Purified full-length human TDG (0.4 μM) was incubated with PARP1 (0.4 μM) and HPF1 (2 μM) in the presence of [$^{32}$P]-NAD$^+$, double-stranded DNA oligomer (200 nM) and U containing DNA oligonucleotides (400 nM) were indicated. Reaction was stopped with 4× SDS loading buffer 0.2 M Tris pH 8.0, 0.4 M DTT, 277 mM SDS, 4.3 M glycerol by heating at 99 °C, 1 min.

## PARylation repair complex dissociation assays

AP-site or SSB containing double-stranded DNA oligonucleotide substrate, generated by digestion of an uracil containing biotinylated duplex oligonucleotide with UDG or UDG and APE1 respectively, 5 min at 37 °C in PARylation buffer. 0.5 μM of the substrate DNA was incubated with human TDG, POLβ, APE1, XRCC1, LIG BER protein (each 0.5 μM), UDG (5 units, New England BioLabs) for 10 min at 30 °C in PARylation buffer (50 mM Tris pH 8.0, 0.5 mg/ml BSA, 1 mM DTT, 2 mM MgCl$_2$) in presence or absence of NAD$^+$ (1 mM), for 15 min at 37 °C. Subsequently equimolar human PARP1 or catalytic inactive PARP1 (PARP1dead) and HPF1 (0.2 μM) were added and incubated for 15 min at 30 °C, shaking 750 rpm. Pre-equilibrated 20 μl magnetic streptavidin beads were added per reaction and incubated for 20 min at 20 °C, shaking 750 rpm. Flow (F) was collected. After washing three times with 250 μl wash buffer (50 mM Tris pH 8.0, 1 mM DTT, MgCl$_2$ 2 mM, 150 mM NaCl; (or 50 mM Tris pH 8.0, 1 mM DTT, 2 mM MgCl$_2$, 20 mM NaCl = low salt wash buffer) at 4 °C, 10 μl of 2× SDS loading dye was added, the samples incubated at 99 °C. Released proteins (elution, E) were separated on 15%-4% SDS−polyacrylamide gels (Mini Protean TGX, BIO-RAD precast gels) and transferred to nitrocellulose membranes (Protran, Amersham) by electroblotting. Blots were then incubated with rabbit polyclonal anti-TDG 141 antibody (raised against recombinant full-length hTDG), dilution 1:20,000, rabbit polyclonal anti-XRCC1 antibody (Sigma-Aldrich, X0629), dilution 1:1000, rabbit polyclonal anti-POLβ antibody (Acris, AM00275PU-N), dilution 1:1000, polyclonal rabbit anti-PAR (Trevigen 4336-BPC-100), dilution 1:1000, polyclonal mouse anti-LIG3 (Genetex (6G9) GTX70147), dilution 1:1000, mouse anti-APEX1 (Invitrogen MA1-440 (13B 8E5C2)), dilution 1:1000 (mouse). rabbit anti-PARP antibody (ab32138 Abcam), dilution 1:1000. All antibodies were diluted in non-fat dry milk TBS (100 mM Tris−HCl pH 8, 150 mM NaCl), 0.1% Tween-20 (Sigma). Analysis was done by chemiluminescence detection (WesternBright ECL, Advansta) on film (FujiFilm) or PXi imaging system, Syngene.

## Plasmids and mutant XRCC1 generation

Construction of the overexpression plasmids of human full-length his-tagged TDG and TDG domains, his-tagged POLβ, APE1, XRCC1, LIG3 were generated by PCR cDNA amplification with adaptor oligonucleotides fused to suitable restriction sites[2,24,29].

hXRCC1-GFP was created by A. Jacobs in the Schär laboratory, using a CAG promoter controlling the hXRCC1 CDS followed by 22 AA linker and eGFP (pCAIP-XRCC1, Addgene # 206032). PAR deficient (pd, Addgene # 206033 and 206036) mutants were designed according to described PARylation sites[11] with following amino acid alterations: S103A, S183A, R186A, S193A, S219A, S220A, S236A, S268A. Reported PARylation sites S234 and S259 were omitted as not to interfere with their modification by phosphorylation. The mutated sequence was synthesized and ordered with Twist Bioscience and integrated into the pCAIP vector using NheI and EcoNI.

## Protein purification

Human BER proteins, POLβ, LIG3, XRCC1, TDG, TDG domains, APE1, PARP1, PARP1dead were expressed and purified as described in[24,29,59]. The protein expression plasmids were electroporated into E. coli BL21(DE3) cells. Starter cultures were grown overnight and diluted with LB broth medium to OD600 of 0.1 and grown at 30 °C to an OD600 of 0.8 under selection with 100 mg/L of ampicillin. Protein expression was induced by addition of IPTG. TDG, XRCC1 and LIG3: 250 μM IPTG, 25 °C for 4 h, APE1: 500 μM IPTG, 25 °C for 6 h, POLβ: 250 μM IPTG, 25 °C for 3.5 h. After harvesting the cells by centrifugation (GSA, Sorvall, 4000 rcf, 30 min, 4 °C), the pellet was resuspended in lysis buffer (50 mM Na-phosphate buffer pH 7.5, 300 mM NaCl, 20% glycerol, 0.1% Tween-20, 1 mM dithiothreitol (DTT), 1 mM phenylmethylsulfonyl fluoride (PMSF)) and sonicated (Bioruptor, Diagenode) to obtain protein fractions. Pre-cleared 6His tagged BER proteins (centrifuged at >30,000 rcf, 60 min, 4 °C) were loaded onto a disposable 15 ml column (Bio-RAD) packed with Ni$^{2+}$-nitrilotriacetic acid (Ni-NTA)-agarose (Qiagen) and washed with 100 ml lysis buffer containing 30 mM imidazole. Elution of bound proteins was performed with 5 ml elution buffer (lysis buffer with 300-500 mM imidazole), and proteins dialyzed and loaded on columns using Akta Explorer 10 (GE Healthcare) HPLC system for further purification as indicated. TDG full size was dialyzed (50 mM Na-phosphate pH 8.0, 50 mM NaCl, 20% glycerol, 1 mM DTT, 1 mM PMSF) and loaded on a 5 ml HiTrap Heparin HP column (GE Healthcare), flow rate 1 ml min$^{-1}$, washed with 10 ml dialysis buffer and the bound proteins eluted with a gradient of 50-800 mM NaCl in 50 ml. After dialysis (50 mM Na-phosphate pH 8.5, 20 mM NaCl, 10% glycerol, 1 mM DTT, 1 mM PMSF) proteins were loaded (flow rate of 1 ml min−1) on a 1 ml HiTrap Q HP (GE Healthcare) column, followed by washing (10 ml dialysis buffer). Bound proteins were eluted using a gradient of 20-500 mM NaCl in 20 ml and homogeneous protein fractions were pooled, dialyzed (50 mM Na-phosphate pH 8.0, 50 mM NaCl, 10% glycerol, 1 mM DTT, 1 mM PMSF) and snap frozen in liquid nitrogen. TDG and TDG domains were further purified by anion exchange chromatography (RESOURCE Q and S GE Healthcare; 20 mM Tris-HCl pH 8.5, 5% glycerol, 1 mM DTT, 0.1% PMSF, 0.005 M NaCl) and eluted with a linear gradient of 0.005−1 M NaCl. Ni-NTA purified APE1 and POLβ were further purified by cation exchange chromatography (RESOURCE S, GE Helathcare; 25 mM Na-phosphate pH 6.9, 5% glycerol, 1 mM DTT, 0.1% PMSF, 0.005 M NaCl) and eluted with a linear gradient of 0.005−1 M NaCl. Ni-NTA purified XRCC1 was purified using Heparin (HiTrap Heparin HP; 25 mM Na-phosphate pH 7, 0.02−1.5 M NaCl, 5% glycerol, 1 mM DTT, 0.1% PMSF) and an anion exchange chromatography (RESOURCE Q; 50 mM Bicine-NaOH pH 8.8, 20% glycerol, 1 mM DTT, 0.1% PMSF, 0.025−1 M NaCl). SUMOylated TDG was generated according to[2] and the SUMO conjugates purified by affinity chromatography by 6His tag purification over Ni$^{2+}$-nitrilotriacetic acid (Ni-NTA)-agarose (Qiagen) and subsequently by fast protein liquid chromatography (FPLC) using

a 1 ml HiTrap Q HP column (GE Healthcare) (50 mM Na-phosphate pH 8, 5% glycerol, 1 mM DTT, 0.1% PMSF, 0.005-1 M NaCl). Pure homogeneous protein fractions were pooled and snap frozen for storage at −80 °C.

### BER reconstitution

Reconstitution of BER reactions were carried out in 20 µl reaction volumes in reaction buffer (50 mM Tris−HCl pH 8.9, 1 mM DTT, 0.1 mg/ml BSA, 1 mM ATP, 2 mM MgCl$_2$, 200 µM dCTP) with TDG (1 pmol), APE (20 fmol, 200 fmol), POLβ (0.1 pmol, 1 pmol), XRCC1 (1 pmol), LigIII (1 pmol) with DNA substrates as indicated. (2 pmol; upper strand 5′-TAGACATTGCCCTCGACGACCCGCCGCCGCGCXGGCCACCCGCACCTAGACGAATTCCG-3′ where X = C was annealed to 1 pmol lower strand 5′-CGGAATTCGTCTAGGTGCGGGTGGCXGGCGCGGCGGCGGGTCGTCGAGGGCAATGTCTA-3′ where X = C, caC, U) as indicated. Reactions were incubated at 37 °C for 1, 15, 20, 30 min as indicated, and the DNA ethanol precipitated O/N at −20 °C. DNA was resuspended and digested with HpaII (1 U, NEB) in 1× CutSmart buffer (NEB) at 37 °C for 1 h. Then DNA was ethanol precipitated O/N at −20 °C and resuspended on glycerol loading buffer (0.5× TBE, 50% glycerol). After separation on an 8% native PAGE, the fluorescence labelled DNA was detected using the blue fluorescence mode of the Typhoon 9400 (GE Healthcare) or PXi imaging system, Syngene and analyzed quantitatively by Fiji ImageJ.

### Nuclear fractionation

A simplified nuclear fractionation was performed according to reference[2]. Briefly, ~20 mio cells were incubated in 200 µl buffer A (10 mM HEPES, pH 7.9, 10 mM KCl, 1.5 mM MgCl$_2$, 0.34 M sucrose, 10% glycerol, 1 mM DTT, 1× cOmplete Protease Inhibitor Cocktail (Roche), 25 nM Talazoparib, 25 nM PDD, 20 mM N-ethylmaleimide), including 0.1% Triton X-100, for 6 min on ice to separate the cytoplasm, which was precipitated with TCA (Supplementary Fig. 4a). After a washing step with buffer A, nuclei were resuspended in buffer B (3 mM EDTA, 0.2 mM EGTA, 1 mM DTT, 1× cOmplete Protease Inhibitor Cocktail (Roche), 25 nM Talazoparib, 25 nM PDD) and incubated for 30 min on ice to separate the nuclear soluble fraction in the supernatant from the chromatin fraction. Total nuclear, nuclear soluble and chromatin fraction were collected in Lämmli buffer and boiled at 95 °C and sonicated for 5 min (30″on/30″off) with a BioRuptor (Diagenode). Proteins were separated using 4−15% Mini-PROTEAN® TGX™ Precast Protein Gels (Bio-Rad) and blotted onto nitrocellulose membranes.

### Single-strand break detection and sequencing

The detection of endogenous SSBs was performed based on the protocol[45]. Genomic DNA was extracted with the Genomic-Tip 100/G Kit (Qiagen) from freshly treated mESC. 50 µg of gDNA were dissolved in a total of 145 µl solution with 15 µl Pol I buffer (10×, NEB buffer) and 15 µl of nick labelling mix containing (10×, 20 µM DIG-dUTPs, 200 µM dATP/dCTP/dGTP, 117 µM ddATP/ddCTP/ddGTP). The mixture was incubated with 5 µl of *E. coli* DNA Pol I (NEB) for 1 min at 16 °C, sample by sample and immediately quenched by the addition of EDTA to a final concentration of 50 µM, and put on ice. The labelled DNA was purified by a precipitation using 2.5 M ammonium acetate and two volumes of 100% EtOH. The DNA was resuspended in H$_2$O and fragmented with a Bioruptor Plus (Diagenode) for 10 cycles of 30 s on/off following a restriction digest with 20 U of each EcoRI, HindIII and XbaI overnight. The fragmented DNA was again purified using DNA clean & concentrator kit (Zymo) according to the manual. Equal amounts of precleared DNA were subjected to immunoprecipitation over night with 2 µg of anti-DIG antibody (Roche) at 4 °C and the immune complexes were recovered with 40 µl of preblocked (tRNA, BSA) Protein G Dynabeads (Invitrogen) for 2 h at 4 °C. The beads were washed once with PBS, three times with NP-40 Buffer (20 mM Tris-HCl, pH = 8.0, 137 mM NaCl, 10% Glycerol, 2 mM EDTA, 1% NP-40), and twice with TE (10 mM Tris-HCl, pH = 8.0, 1 mM EDTA). The immunocomplexes were eluted twice with 50 µl of TE with 0.5% SDS prior to digestion with 50 µg/ml Proteinase K at 52 °C for 2 h. The DNA was then purified with the ChIP Clean & Concentrator Kit (Zymo) and subjected to library preparation.

### Library preparation and high-throughput sequencing

Libraries of Input and IP samples (10 ng DNA each) were prepared using the KAPA HyperPrep Kit (Roche) following the manufacturers protocol. Subsequent paired-end sequencing (75 cycles) was performed on an Illumina MiSeq system at the genomics facility Basel to an average depth of 50 mio reads per sample.

### Bioinformatic processing

Reads where aligned to the mouse genome (mm10 UCSC version) with bowtie2 (version 2.3.4.2) and extra options "--maxins 2000 --no-mixed --no-discordant --local --mm". Duplicates were marked with picard tools (version 2.9.2) and resulting BAM files were filtered by removing reads falling into ENCODE blacklist regions (version 2014 + manual removal of high coverage regions) using Rsamtools (R version 3.6, Bioconductor version 3.10).

Peaks were called across all replicates of a sample group using HOMER (version 4.11) and "findPeaks" with specifying commands "-style histone - with the corresponding input as a control. Signals of SSB outside of significant enrichment (peaks) were counted as log2 fold enrichment of IP over Input by using "bamCount" (bamsignals package V1.22.0) at genomic ranges of interest (V1.46.0), followed by normalization by trimmed mean of M-values according to the library size using edgeR (V3.13). Graphs were created using R-studio version 4.0.3 and the R Package "ChIPseeker" (V1.26).

### Total RNA-seq analysis

TotalRNASeq reads were mapped to mm10 mouse genome using STAR (version 2.6). Gene expression values were extracted using HTSeq (version 0.12.4) taking bam files (sorted by Coord) generated by STAR aligner. HTSeq output files of replicates were used as input files to compute differential gene expression using the generalized linear model implemented in the Bioconductor package DESeq2 (version 1.26.0).

### Mass spectrometry analysis of 5-methylcytosine and oxidized derivatives

DNA was extracted and purified with the Genomic tip 100 G Kit (Qiagen). Ultra-high-performance liquid chromatography-tandem mass spectrometry analysis was performed on 10 µg of genomic DNA digested to nucleosides in 10 mM ammonium acetate buffer pH 6.0, 1 mM MgCl2 for 60 min at 40 °C, with nuclease P1 (Sigma, N8630), benzonase (Santa Cruz Biotech, sc-202391) and alkaline phosphatase (Sigma, P5931). Digested samples were precipitated with 3 volumes of acetonitrile and supernatants were lyophilized and dissolved in a solution of internal standards (I.S.) for analysis. An Agilent 1290 Infinity II UHPLC system with an ZORBAX RRHD Eclipse Plus C18 150 × 2.1 mm (1.8 µm) column protected with a ZORBAX RRHD Eclipse Plus C18 5 × 2.1 mm (1.8 µm) guard column (Agilent) was used for chromatographic separation. The mobile phase consisted of A: water and B: methanol (both added 0.1% formic acid), for 5hm(dC), 5f(dC) and 5ca(dC) starting at 0.15 ml/min flow of 5% B for 0.5 min followed by 3.5 min gradient of 5−15% B, 3 min of 15−90% B while increasing the flow to 0.22 ml/min, 0.5 min of 90% B, and 4 min re-equilibration with 5% B. Unmodified nucleosides and 5m(dC) were chromatographed at 0.22 ml/min with a 4 min gradient 5-95% B and 4 min re-equilibration with 5% B. Mass spectrometric detection was performed using an Agilent 6495 Triple Quadrupole system operating in positive electrospray ionization mode. The following mass transitions were monitored: 242.1/126.1 (5 m(dC)); 258.1/142.1 (5hm(dC)); 256.1/140.1

(5 f(dC)); 272.1/156.1 (5ca(dC)); 252.1/136.1 (dA); 228.1/112.1 (dC); 268.1/152.1 (dG); 243.1/127.1 (dT); 257.1/136.1 ($^{13}C_5$-dA I.S.); 246.1/130.1 ($^{13}C,^{15}N_2$-dT I.S.); 245.1/129.1 ($d_3$−5m(dC) I.S.); 261.1/145.1 ($d_3$−5hm(dC), I.S.), 264.1/112.1 (gemcitabine, a dC analog coeluting with 5ca(dC) and used as I.S. for 5 f(dC) and 5ca(dC)).

## Reporting summary

Further information on research design is available in the Nature Portfolio Reporting Summary linked to this article.

## Data availability

The SSB-seq and RNA-Seq data have been deposited in NCBI's Gene Expression Omnibus and are accessible through GEO Series accession number GSE166963 and the BioProject accession number PRJNA743896, respectively. Source Data is provided for this work. Source data are provided in this paper.

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

## Acknowledgements

We thank Dr. Florian Geier, DBM Bioinformatics Core Facility Department of Biomedicine, University of Basel for his support with the analysis of the SSB-seq. We also want to thank Thimo Müller from the Primo Schär group for providing a R-script. We thank Dr. Pascal Lorenz and the Microscopy core facility for their kind assistance and flexibility. We also want to thank Dr. Gaël Auray from the DBM FACS core facility for his help. Calculations were performed at sciCORE (http://scicore.unibas.ch/) scientific computing core facility at the University of Basel. The work was supported by the Swiss National Science Foundation (SNSF_156467, 182280) (PS) and the Walter Honegger Foundation, Zurich (R.S.). J.X. was supported by the Mäxi-Foundation, K.G. by the Swiss Cancer Research Foundation (KFS-3740-08-2015). Research in the laboratory of MOH was funded by the Canton of Zurich and the Swiss National Science Foundation Grants (SNF 31003A_176177 and 310030_205202). The Proteomics and Modomics Experimental Core facility (PROMEC) is funded by the Norwegian University of Science and Technology (NTNU) and The Central Norway Regional Health Authority. This facility is a member of the National Network of Advanced Proteomics Infrastructure (NAPI), which is funded by the Research Council of Norway INFRASTRUKTUR-program (project number: 295910) (G.S., C.V.).

## Author contributions

Conceptualization: R.S.; Methodology: R.S., S.D.S., D.S. J.X., K.G., E.F., M.O.H., P.S., C.V., G.S.; Investigation: R.S., S.D.S., J.X., K.G., E.F., D.S., C.V.; Visualization: R.S., K.G., S.D.S., D.S.; Funding acquisition: R.S., P.S., M.O.H.; Project administration: R.S.; Supervision: R.S.; Writing – original draft: R.S., S.D.S.; Writing – review & editing: R.S., S.D.S., J.X., K.G., D.S., E.F., M.O.H., P.S.

## Competing interests

The authors declare no competing interests.
