## [Peer Review File · Nature Communications]

Covalent PARylation of DNA base excision repair proteins regulates DNA demethylationREVIEWER COMMENTS

Reviewer #1 (Remarks to the Author):

The role of PARP1 and other DNA damage-activated PARPs in the DNA damage response has been studied for many years. Most of these studies have focused on the transient automodification of PARP1 following activation by DNA damage. The prevailing model from these studies is that proteins involved in the repair of DNA single strand breaks and also possibly base lesions are recruited to sites of damage by binding to PARylated PARP1 that is released from the DNA damage site but presumably remains in the vicinity. While it has been known that other proteins, such as histones and DNA repair proteins, are PARylated, not much is known about the biological consequences of this covalent modification. In this study the authors provide intriguing preliminary evidence that PARylation of proteins involved in the removal of 5 methyl cytosine residues by BER promotes their release from DNA and likely has a similar function in the repair of DNA damage. This is potentially paradigm changing shift in how we view the role of covalent parylation of DNA repair proteins. There are, however, a number of concerns;

(i) While the study utilizes talazoparib, a cytotoxic PARP trapper, in cell-based studies, PARP1 null mice develop relatively normally raises questions about the biological significance of this proposed mechanism in terms of demethylation that could be addressed experimentally or at least discussed.

(ii) There are some inconsistencies in the model. If PARP1 is activated by an AP site of SSB, the proposed subsequent parylation of proteins in the BERosome may result in dissociation of the complex prior to DNA synthesis and ligation. Does parylation differentially impact different steps of the BER reaction?

(iii) The majority of studies examine either enzymatic processing of a DNA substrate or binding to the DNA substrate and are primarily qualitative in nature. It is not evident how much of a specific protein is PARylated. In addition, the effect of PARylation on specific proteins should be examined by making mutant versions that are non-PARylatable. The effect of PARylation on DNA binding could be quantitated by SPR or single molecule approaches.

Minor comments.

HPF1 is included in the biochemical assays but there is no discussion about the effect of PARylation by HPF1

The PARP1 dead mutant is not described- is this defective for catalytic activity and/or DNA binding?

Reviewer #2 (Remarks to the Author):

Tang et al use biochemical reconstitution experiments to show that PARP1 binds to abasic sites and SSB generated through TET-TDG mediated demethylation and covalently PARylates Base excision repair factors TDG, APE1, XRCC1, Pol β and Lig3. While non-covalent interaction of BER factors with PARylated PARP1 serves as a recruitment signal to sites of DNA damage, covalent PAR-modification results in the release of BER factors from DNA. The authors state that this allows timely dissociation of the BER complex from repair sites to enable protein turnover and efficient completion of DNA repair. Although PARylation of BER has been reported previously in several high-throughput screens, the exact function of this modification has not been described previously. Therefore, these are interesting and novel insights into the mechanism of BER and how it can be controlled on different levels via PARP1 and PARylation. The authors propose an active involvement of PARP1 in regulating demethylation through PARylation of TDG in mESCs. The authors should include more data to show the direct involvement and molecular mechanism of covalent modifications of BER proteins in cells for active demethylation and discuss how the different modes and outcome of PARylation are controlled.

Points that need to be addressed:

Using biochemical reconstitution assays the authors nicely demonstrate that covalent PARylation of BER proteins results in their dissociation from DNA. This indicates that release of BER proteins from repair sites requires PARylation and enables efficient repair. While the reconstitution assays give a very clear and important insight into this mechanism, this should be also tested in cells, by looking at recruitment and dissociation of BER proteins in absence and presence of PARylation. As PARylation of PARP1 is also required for efficient recruitment of BER proteins, PARP Inhibitors can properly not be used in this case. However, the authors could for example generate BER protein mutants that cannot be PARylated and check their repair capacity and binding/dissociation at DNA damage sites in cells.

The authors show that SSB levels at potential DNA demethylation sites in mESCs are reduced upon PARP inhibition. The effects are rather mild though with a reduction from 24 % to 18% at sites of CaC excision and 37% to 29% at dynamically methylated CpGs upon talazoparib treatment. Why is the reduction not more profound? What about the SSBs that are still present after Talazoparib treatment (and TDG knockout)? Are they mediated through other TDG-independent mechanisms? To show that PARP1 activity is indeed required for active demethylation, the authors need to analyse methylation levels in mESCs upon PARP1 inhibition and/or TDG depletion. In addition, rescue experiments with WT TDG or a TDG mutant that cannot be PARylated could be performed to show that PARylation of TDG is required for active demethylation in mESCs.

How is SUMOylation of TDG coordinated with PARylation and which of these two modifications is required for the release of TDG from the demethylation sites during repair?

The proposed model suggests that PARP1 senses AP sites and SSBs generated by TET-TDG at sites of active demethylation, is activated and then PARylates TDG leading to its release from DNA. Inhibition of PARP leads to reduced TDG turnover, less excision and therefore fewer SSB. Inhibition of PARP1-mediated turnover of general BER factors at other SSBs and blocking of PAR signaling for the recruitment of BER factors, outside of demethylated areas, results in more spontaneous AP sites and SSBs. It is not clear though how this distinction at the two different repair sites could be achieved. Are other general BER factors besides TDG at demethylation sites not PARylated and is automodification of PARP1 not required here for BER factor recruitment? Is TDG at spontaneous SSBs or AP sites outside of areas of active demethylation not PARylated? How would this be controlled and how would PARP1 distinguish between these different repair scenarios? The authors should address this with more experiments and expand the discussion.

It would be interesting to repeat the experiment from Figure 4 with a PARG Inhibitor or PARG depletion, to test what effect excessive PARylation has on demethylation and SBB repair in mESCs.

Minor Points:

- The claim in the title of Figure legend 1 that "Covalent PARylation of BER proteins stimulates DNA demethylation" is quite strong. In Figure 1 it is not directly shown that demethylation of DNA is directly affected by PARylation of BER proteins. A later product of the demethylation reaction, G:caC, is used in the biochemical assay in Figure 1 d and not methylated DNA. The title should be changed.
- Figure quality needs to be improved. Font sizes are too small and schematics are in small size and low quality and thus difficult to read. Maybe this is due to the PDF compression during the upload of the manuscript?
- Figure legend 1 F: Lig3 written twice
- Figure 4 d: No legend for blue and red curve. Does the red curve correspond to Talazoparib treated cells? Is there a general increase in SSB outside the high TERT activity site?

Reviewer #3 (Remarks to the Author):

In this study, Xu et al. report that poly(ADP-ribosyl)ation facilitates TET-TDG-mediated DNA demethylation via regulating base excision repair (BER) pathway. TDG is a DNA glycosylase and recognizes 5fC and 5csC. It mediates the removal of 5fC and 5csC from genomic DNA. Via BER pathway, unmodified cytosine is incorporated into genomic DNA to complete the active demethylation process. Here, the authors show that PARP1 PARylates a number of BER factors, such as TDG, for promoting dissociation of BER factors from genomic DNA upon repairing lesions. Overall, the manuscript provides an interesting hypothesis but lacks compelling evidence to support the major conclusions.

Specific points:

1. It is not surprising to see that many DNA damage repair factors are poly(ADP-ribosyl)ated, particularly in an in vitro PARylation setting, as PARylation can be catalyzed on several amino acid residues. Numerous PARylation events occurs under extreme genotoxic stress conditions, and these events may be just passenger events. The key question that needs to be addressed is the biological function(s) of PARylation during DNA demethylation in mESCs. The authors used PARP inhibitor Talazoparib to examine the functions of PARylation, which might not be suitable, as this PARP inhibitor shows strong PARP1 trapping activities. Once PARP1 is trapped on genomic DNA, it will generate other secondary lesions, such as DNA double-strand breaks. Thus, it would be better to use PARP1 KO cells to examine the biological significance of PARP1-mediated PARylation.
2. The working model proposed in this manuscript is that PARylation mediates the release of BER factors perhaps during intermediate steps, thus facilitates BER. However, it would be straightforward to examine if BER factors are retained at DNA lesion in PARP1 KO mESCs.
3. It may be strange to see that XRCC1 and POL beta bind to nicked DNA in the presence of PARP1 dead since PARP1 dead also binds to nicked DNA and may compete with XRCC1 and POL beta to interact with DNA.
4. The font in figures is too small.

POINT-TO-POINT RESPONSE TO REVIEWERS' COMMENTS

We thank all Reviewers for giving our work careful attention and for appreciating the novelty of the finding that covalent PARylation coordinates molecular processes in DNA repair. Their competent, constructive and insightful criticism and input have guided us in significantly improving the clarity and quality of the manuscript. All reviewers' questions and respective revisions in the manuscript are addressed and outlined in the following.

Reviewer #1 (Remarks to the Author)

The role of PARP1 and other DNA damage-activated PARPs in the DNA damage response has been studied for many years. Most of these studies have focused on the transient automodification of PARP1 following activation by DNA damage. The prevailing model from these studies is that proteins involved in the repair of DNA single strand breaks and also possibly base lesions are recruited to sites of damage by binding to PARylated PARP1 that is released from the DNA damage site but presumably remains in the vicinity. While it has been known that other proteins, such as histones and DNA repair proteins, are PARylated, not much is known about the biological consequences of this covalent modification. In this study the authors provide intriguing preliminary evidence that PARylation of proteins involved in the removal of 5 methyl cytosine residues by BER promotes their release from DNA and likely has a similar function in the repair of DNA damage. This is potentially paradigm changing shift in how we view the role of covalent parylation of DNA repair proteins. There are, however, a number of concerns.

(i) While the study utilizes talazoparib, a cytotoxic PARP trapper, in cell-based studies, PARP1 null mice develop relatively normally raises questions about the biological significance of this proposed mechanism in terms of demethylation that could be addressed experimentally or at least discussed.

Response:

Indeed, PARP1 null mice are viable and healthy, but the functional defects of protein inhibition are not comparable with genetic depletion and can reveal aspects of dominant negative interference with molecular pathways. PARP1 is involved in many repair processes (Ray Chaudhuri & Nussenzweig, 2017), which also do not display major defects in the absence of PARP1. This is in line with the published results that showed a redundancy between the members of PARP proteins and compensatory pathways, which are activated when PARP is genetically depleted. It is also noteworthy, that the core enzymatic functionalities in BER, do not require PARP per se (e.g. Weber et al., 2016), instead PARylation seems to rather fine tune the molecular transaction involved in the context of chromatin to optimize the robustness, accuracy and efficiency of the essential BER process, similar to what was reported for SUMOylation (Steinacher et al., 2019).

We relate to the reviewer's concerns and therefore performed additional experiments to further test the biological significance of PARylation in ESCs. We now demonstrate that hyper-PARylation in cells by inhibition of PARG inhibition results in increased levels of fC and caC DNA demethylation intermediates. This we show in the newly added Figure 1c. Moreover, we show that hyper-PARylation leads to reduced chromatin-association of BER factors. This result has been added as Figure 2g to the revised manuscript. Further we show that recruitment dynamics of XRCC1 to induced DNA damage is modulated by PARylation. This is shown in Figures 3d-f and Figure S3 f-k in the revised manuscript. Overall, these new added data clearly show that covalent PARylation provides an important mechanism to orchestrate the dynamics of active DNA demethylation and DNA break repair in ESCs.

(ii) There are some inconsistencies in the model. If PARP1 is activated by an AP site of SSB, the proposed subsequent parylation of proteins in the BERosome may result in dissociation of the complex prior to DNA synthesis and ligation. Does parylation differentially impact different steps of the BER reaction?

Response: This is a very interesting question and right to the point. We definitely agree with the reviewer that PARylation may differentially affect individual steps of BER, starting with DNA damage recognition and recruitment of BER and SSB factors and ending with the actual repair process. We showed previously that the core BER / SSB proteins assemble at sites of DNA base excision as a multiprotein complex (BERosome, Steinacher et al., 2019) which then releases TDG by SUMOylation. As this is a fast and robust process, bringing in all enzymatic activities required for AP-site repair at once, we reason that AP-site incision, repair synthesis and ligation, and PARP1 activation occur nearly simultaneously with AP-site repair being completed prior to full PARylation of BER proteins. This is consistent with PARylation affecting the DNA binding affinity but not the enzymatic activities of BER proteins. Also PARG was shown to be recruited to a DNA lesion with equal dynamics as PARP1 (Aleksandrov et al., 2018), implicating a tight control of PARylation and de-PARylation events. This may provide windows of opportunity and a niche for «BER» at sites of DNA lesions, facilitating proper repair and turnover of BER factors. Also, PARP activation goes hand in hand with its autoPARylation, which efficiently dissociates PARP1 itself from the DNA (Krüger et al., 2020), hence limiting its activity once activated. We addressed this in the discussion and highlighted the text.

Another situation of differential impact of PARylation on BER/SSB arises at the initial recruitment step, where non-covalent PAR interactions play a significant role. Differentiating the impact of non-covalent PARylation and covalent PAR interactions in this context will be an important next step and a challenging one as well and is not the scope of this work. The scope of this work is to first establish a function of covalent PARylation in coordinating a multienzyme DNA repair process. Notably, in the LiDD experiments (Figure 3 d-f), which we performed for the revision, we observed that hyper-PARylation after PARGi leads to a slightly longer retention of XRCC1 at the site of damage when compared to DMSO treated cells (Figure 3e vs. Figure S3j). This indicates that although BER proteins engage, and therefore will likely be PARylated, the non-covalent PAR interactions that initially recruited the BER protein, still keep XRCC1 in place. Thus, PARylation of BER proteins and non-covalent interactions of BER proteins with PAR are likely to fine-tune together the overall affinity of BER proteins to sites of DNA damage within chromatin. We added the results of the LiDD experiment to the manuscript and highlighted the results in the text.

Although mechanistic details will need to be resolved, the general finding of a function of covalent protein PARylation in dissociation of DNA repair complexes is shown in different ways and well-established by our data. We therefore prefer to explain the data in a simple model, focusing on PARylation only, to illustrate this conceptually novel function in the context of DNA methylation (Figure 4j). Nevertheless, other posttranslational modifications are likely to play a coordinating role in BER / SSB as well. For instance, PARylation was shown to promote SUMOylation of XRCC1, which is critical for BERosome formation and coordination of DNA demethylation. The formation of the BERosome on the other hand is likely to affect the PARylation of the individual BER proteins. We added in the discussion a section where we refer to the observation that PARylation may promote the recruitment of the SUMO ligase TOPORS, which ultimately may facilitate BERosome formation. SUMOylation and PARylation might both be part of a regulatory network to control the transactions of the BER complex through the individual steps during SSB and BER mediated DNA demethylation. Disentangling this is not within the scope of this paper.

(iii) The majority of studies examine either enzymatic processing of a DNA substrate or binding to the DNA substrate and are primarily qualitative in nature. It is not evident how much of a specific protein is PARylated. In addition, the effect of PARylation on specific proteins should be examined by making

mutant versions that are non-PARylatable. The effect of PARylation on DNA binding could be quantitated by SPR or single molecule approaches.

Response: This a valid and very important question, relevant for many posttranslational protein modification studies out there, most of which show at most approximately quantitative data. Quantitation of the fraction of a PARylated protein is a challenge both *in vitro* and (even more) *in vivo* due the intrinsic heterogeneity of the modification (i.e. from mono – ADP ribosylation to complex branched poly-ADP ribosylation). We estimate that, in our biochemical assays, about 30-40% of the protein molecules were PARylated, based on the densitometric quantitations provided. We are fully aware of the value of precise quantitation in the interpretation of enzymatic activities measured in biochemical assays and we are working towards more precise assessments in our efforts to investigate the functional crosstalk post-translational modifications in BER. We do believe though that the level of quantitation proved in the manuscript is sufficient to support the conclusions drawn.

Regarding the generation of non-PARylatable mutants, we agree with the reviewer that this would be a most convincing control. Yet again, this is challenging for many reasons (unknown PARylation sites, multiple predicted sites in most BER enzymes, secondary effects of point mutations). Yet, we made an effort to provide such data, at least for one protein, and now provide experimental data for a PARylation deficient variant of hXRCC1. We have chosen XRCC1 because of its central scaffold function in BER and SSB and exchanged eight out of ten reported PARylation sites for alanine; to avoid potential effects on hXRCC1 phosphorylation we left the two known phosphorylation/PARylation sites intact. The mutant hXRCC1 (and the wildtype as control) was expressed in U2OS cells and examined for DNA damage association. The PARylation deficient mutant re-associated significantly more efficiently to microlaser-induced DNA damage than the wildtype protein under conditions where pre-PARylation of the wildtype XRCC1 was induced by a first wave of DNA damage and stabilized with PARG inhibition. The data show that PARylated XRCC1 (stabilized by PARG inhibition) has a lower affinity to nuclear DNA damage than a non-PARylated XRCC1. These new data are now presented in Figure 3c-f, Figure S3f-k and discussed in the revised manuscript.

Minor comments.

HPF1 is included in the biochemical assays but there is no discussion about the effect of PARylation by HPF1. The PARP1 dead mutant is not described - is this defective for catalytic activity and/or DNA binding?

Responses: Thank you for pointing out this omission. We used HPF1 in our biochemical assays as a co-factor of PARP1 to increase PARylation specificity and to facilitate detection of covalent protein PARylation based on published evidence (Suskiewicz et al., 2020). This is now explained in the main text and the reference is included in the methods.

The PARP1 dead version is defective in its catalytic activity but still able to bind DNA. This is explained more clearly in the revised text and is appropriately referenced.

Reviewer #2 (Remarks to the Author)

Tang et al use biochemical reconstitution experiments to show that PARP1 binds to abasic sites and SSB generated through TET-TDG mediated demethylation and covalently PARylates Base excision repair factors TDG, APE1, XRCC1, Pol β and Lig3. While non-covalent interaction of BER factors with PARylated PARP1 serves as a recruitment signal to sites of DNA damage, covalent PAR-modification results in the release of BER factors from DNA. The authors state that this allows timely dissociation of the BER complex from repair sites to enable protein turnover and efficient completion of DNA repair. Although PARylation of BER has been reported previously in several high-throughput screens, the

exact function of this modification has not been described previously. Therefore, these are interesting and novel insights into the mechanism of BER and how it can be controlled on different levels via PARP1 and PARylation. The authors propose an active involvement of PARP1 in regulating demethylation through PARylation of TDG in mESCs. The authors should include more data to show the direct involvement and molecular mechanism of covalent modifications of BER proteins in cells for active demethylation and discuss how the different modes and outcome of PARylation are controlled.

Points that need to be addressed:

Using biochemical reconstitution assays the authors nicely demonstrate that covalent PARylation of BER proteins results in their dissociation from DNA. This indicates that release of BER proteins from repair sites requires PARylation and enables efficient repair. While the reconstitution assays give a very clear and important insight into this mechanism, this should be also tested in cells, by looking at recruitment and dissociation of BER proteins in absence and presence of PARylation. As PARylation of PARP1 is also required for efficient recruitment of BER proteins, PARP Inhibitors can properly not be used in this case. However, the authors could for example generate BER protein mutants that cannot be PARylated and check their repair capacity and binding/dissociation at DNA damage sites in cells.

Response: We are grateful for the Reviewer's insightful comments and questions. We understand and agree with the points concerning the in-cell validation of the mechanistic model proposed mainly on the basis of biochemical in vitro assays. Differentiating the functions of non-covalent and covalent PARylation in BER in cells is a considerable challenge and we made a significant effort to address this important question with additional experiments. We now provide data to document the effects of stimulation of active DNA demethylation (Vitamin C treatment) and PARG inhibition (PARGi) on the levels of PARylation in cells (i.e. ESCs) (Figure 1b). We show that stimulation of active BER mediated DNA demethylation by VitC, as well as PARGi cause hyper-PARylation in ESCs (Figure 1c). Notably, hyper-PARylation after PARGi results in reduced chromatin association of BER factors (Figure 2f, g). As suggested by the reviewer, we generated an hXRCC1 PARylation deficient mutant where we altered 8 out of 10 reported PARylation sites (Hendriks et al., 2019) and introduced it into U2OS cells (Figure 3c, Supplemental Figure S3 f-i). The generated XRCC1 mutant shows reduced PARylation but is still recruited to a laser induced DNA damage with similar kinetics compared to wild type XRCC1 (Figure 3f). Interestingly, hyper-PARylation upon PARGi treatment in U2OS cells delayed the damage association of XRCC1wt-GFP significantly more compared to PARylation deficient XRCC1pd-GFP, which relocated significantly faster (~25%) to a laser induced DNA damage (Figure 3f). These findings support our biochemical results and show that a previously engaged and therefore pre-PARylated hXRCC1, is less likely to re-engage with new damage and requires de-PARylation to regain optimal damage affinity. We describe these important results in our revised manuscript (i.e. highlighted sections).

The authors show that SSB levels at potential DNA demethylation sites in mESCs are reduced upon PARP inhibition. The effects are rather mild though with a reduction from 24 % to 18% at sites of CaC excision and 37% to 29% at dynamically methylated CpGs upon talazoparib treatment. Why is the reduction not more profound? What about the SSBs that are still present after Talazoparib treatment (and TDG knockout)? Are they mediated through other TDG-independent mechanisms? To show that PARP1 activity is indeed required for active demethylation, the authors need to analyse methylation levels in mESCs upon PARP1 inhibition and/or TDG depletion. In addition, rescue experiments with WT TDG or a TDG mutant that cannot be PARylated could be performed to show that PARylation of TDG is required for active demethylation in mESCs.

Response: These questions are very central and to the point. In line with the accepted model of SSBR where PARylation orchestrates the recruitment of repair factors to sites of DNA damage, the

expectation (including ours) for SSB profiling experiment was that the amount of genomic SSBs would increase upon treatment of cells with the PARP inhibitor Tal. In this regard, we see the observed and reproducible reduction of SSBs upon Tal to a lower level than in untreated ESCs as a quite significant effect. Nonetheless, the reduction of SSBs at sites of caC excision and TET-mediated oxidation may not be more profound due to a number of reasons: (i) PARP inhibition by Tal will not only inhibit PARylation of BER proteins but also of the chromatin at sites of damage. This will affect the generation and repair of SSB in a multifactorial manner. (ii) The dynamics of TDG-mediated excision and strand incision by APE1 can be affected by multiple factors in addition to post-translational modifications, such as APE1 itself, or the bifunctional glycosylases NEIL (Schomacher et al., 2016) that were shown to stimulate the turnover of TDG, indicating that covalent PARylation is not the sole regulator of this process and that there may be compensatory mechanisms. Also, we reason that the effect of Tal on BER is not a full inhibition of the process, but a reduction in the rate of dynamic DNA methylation (DNA methylation, active DNA demethylation) through a reduction of PARylation dynamics. (iii) Only about 12% of detected SSB peaks in wildtype ESCs are located in intergenic regions (Supplementary Figure 4a) while the rest are associated with genic regions such as promoters, 5'UTRs or exons etc. Transcriptional activity was previously associated with increased levels of DNA break formation (Baranello et al., 2014; Wu et al., 2021). Upon Tal treatment, we observed a change in the pattern of SSB formation at sites of ongoing active DNA demethylation, with SSBs being less focused in the center of the DNA demethylation activity and more spread out into the neighboring areas (Figure 4c,d). This indicates an activation of alternative, compensatory pathways to maintain the transcriptional output but operating in a less coordinated manner, which finally leads to a reduction of targeted SSBs.

As suggested by the reviewer, we performed additional experiments to demonstrate the biological significance of PARP1 activity for active demethylation. We now show in the revised manuscript in Figure 1b that stimulation of active DNA demethylation by Vitamin C increased PARylation; and in Figure 1c we measure DNA demethylation intermediates and show that PARG inhibition indeed increases levels of oxidized mC-derivates. We described these results in the revised manuscript and highlighted the newly added sections in the text.

We agree that the creation of a non-PARylatable TDG variant might be helpful and informative. However, given that TDGs sole reported PARylation site (S85 (Hendriks et al., 2019)) is at the same time a functional phosphorylation site, mutating this amino acid would generate ambiguous and inconclusive results. We therefore decided not to do this. Also, we detected PARylation within the CORE domain of TDG, downstream of S85 (Figure S1e) at unknown modification sites. This suggests the existence of alternative, unpredictable PARylation sites, that will be very difficult to control. We therefore decided to focus the mutagenesis approach on the XRCC1 protein.

How is SUMOylation of TDG coordinated with PARylation and which of these two modifications is required for the release of TDG from the demethylation sites during repair?

Response: This is indeed an intriguing question that can be extended to even more post-translational modifications of TDG that are present in TDG. Interestingly, it was previously shown that SUMOylation of XRCC1 is stimulated via PARylation and the recruitment of TOPORS (Hu et al. 2018). In this context it is noteworthy that we have demonstrated that SUMOylation of XRCC1 is important for TDG-BERosome assembly and coordination of TDG BER activity during DNA demethylation in differentiating ESCs (Steinacher et al. 2019). It is a very attractive hypothesis that PARylation may modulate XRCC1 SUMOylation to promote BERosome formation. We added a section in the discussion of the revised manuscript to discuss this intriguing concept.

The proposed model suggests that PARP1 senses AP sites and SSBs generated by TET-TDG at sites of active demethylation, is activated and then PARylates TDG leading to its release from DNA. Inhibition of PARP leads to reduced TDG turnover, less excision and therefore fewer SSB. Inhibition of PARP1-mediated turnover of general BER factors at other SSBs and blocking of PAR signaling for the recruitment of BER factors, outside of demethylated areas, results in more spontaneous AP sites and SSBs. It is not clear though how this distinction at the two different repair sites could be achieved. Are other general BER factors besides TDG at demethylation sites not PARylated and is automodification of PARP1 not required here for BER factor recruitment? Is TDG at spontaneous SSBs or AP sites outside of areas of active demethylation not PARylated? How would this be controlled and how would PARP1 distinguish between these different repair scenarios? The authors should address this with more experiments and expand the discussion.

It would be interesting to repeat the experiment from Figure 4 with a PARG Inhibitor or PARG depletion, to test what effect excessive PARylation has on demethylation and SSB repair in mESCs.

Response: Why inhibition of PARylation in ESCs leads to less BER associated SSBs but elevated levels of spontaneous SSBs, is an interesting question. In this context it is noteworthy that there is a fundamental difference between BER versus SSB repair in terms of damage recognition and repair.

In BER, DNA base damage recognition is highly specific and mediated by lesion specific active site pockets of DNA glycosylases, which is rate limiting for repair. We have previously shown that TDG forms a BERosome with XRCC1 and SSB proteins, which couples base excision, AP-site cleavage and SSB repair (Steinacher et al. 2019). Our biochemical data demonstrate that PARylation reduces TDG DNA binding to stimulate enzymatic turnover and thus accelerates base excision. This suggests that TDG PARylation is required for its turnover, and after de-PARylation TDG can bind DNA again and re-engage in new DNA demethylation cycles. As a consequence, in PARP inhibited cells, TDG activity is impaired, resulting in overall reduced DNA demethylation activity and consequently less BER associated AP-sites and SSBs. This is in line with our data, which show that after talazoparib treatment the levels of BER related SSBs are reduced.

In contrast, repair of “spontaneous” or “random” SSBs requires PARP, which senses these SSBs and starts the repair process. It was shown before, and is well accepted, that at sites of “spontaneous” SSBs, PARylation at the site of damage orchestrates the recruitment of XRCC1 and SSB proteins. In line with this, we show that inhibition of PARP (i.e. talazoparib) in ESCs results in accumulation of “spontaneous” SSBs.

Concerning the potential PARylation of TDG at sites outside active DNA demethylation regions: The main, if not the only function of TDG supported by genetic evidence is the initiation of active DNA demethylation following mC oxidation by TET. TDG deficiency in cells and mice does not generate mutator phenotype (tested thoroughly by different methods in different labs (e.g (Cortazar et al., 2011)). We therefore reason that TDG is mostly engaged in and PARylated in the context of active DNA demethylation. Of course, we cannot exclude that there may be situations where TDG is PARylated in other BER scenarios as well, such as in the repair of repair G:T or G:U mismatches resulting from deamination of mC and C, as suggested by its biochemical activity. It may also be PARylated in these situations, but it is more likely that other glycosylases like UNG, SMUG1 and MBD4 would be predominantly engaged in these situations (as indicated by all genetic evidence).

We considered measurements of SSBs in PARG inhibited ESCs but decided against this experiment for the following reason. We found that PARG inhibition and hyper-PARylation in ESCs increases levels of fC and caC, as shown in the newly added Figure 1c. This suggests an inhibitory effect of hyper-PARylation base excision activity, which will result in lower levels of BER related SSBs in regions undergoing active DNA demethylation. On the other hand, we showed that covalent PARylation of all downstream BER factors reduces their repair efficiency, which would in turn predict an increase of

SSB at these sites. Disentangling these and other interfering processes of PARGi treatment on BER will not be possible by using SSB as a readout.

Therefore, we chose a different approach and studied how excessive PARylation affects BER proteins and BER mediated DNA demethylation in cells. In the revised manuscript we now show that PARGi treatment in ESCs results in elevated ADP-ribose levels (Figure 1b) and elevated fC and caC levels (Figure 1c). Further we show that PARGi treatment of ESCs leads to a reduced association of TDG and XRCC1 to chromatin (Figure 2g), and iv) reduced recruitment of XRCC1 to induced DNA damage (Figure 3d-f). Together, these results clearly show that DNA damage and chromatin association of BER proteins and BER is regulated by PARylation in ESCs.

Minor Points:

- The claim in the title of Figure legend 1 that “Covalent PARylation of BER proteins stimulates DNA demethylation” is quite strong. In Figure 1 it is not directly shown that demethylation of DNA is directly affected by PARylation of BER proteins. A later product of the demethylation reaction, G:caC, is used in the biochemical assay in Figure 1 d and not methylated DNA. The title should be changed.

The title of Figure legend 1 was adjusted, accordingly the error in legend 1f was corrected.

- Figure quality needs to be improved. Font sizes are too small and schematics are in small size and low quality and thus difficult to read. Maybe this is due to the PDF compression during the upload of the manuscript?

We improved the quality of the figures and increased font size in schemata and figures to improve readability.

- Figure legend 1 F: Lig3 written twice

Thank you for pointing this out, the Figure legend is now corrected.

- Figure 4 d: No legend for blue and red curve. Does the red curve correspond to Talazoparib treated cells? Is there a general increase in SSB outside the high TERT activity site?

We improved the legend of Figure 4 according to the reviewer's comment by adding the color code of the curves. There is no genome-wide increase of SSBs upon Tal detectable outside of sites displaying TET/TDG activity. Below you can find a plot depicting the ranked read distribution over all chromosomes binned into 1000 bp windows. At the very right are the regions with highest concentrations of reads, which mostly are detected as significant peaks. A genome-wide increase of SSBs upon Tal would be represented by the reddish curves (Tal treated samples) above the blueish ones (DMSO treated samples). This is, however, only the case in regions with very low read counts (bottom left of the curve). Significant increases of SSB signal upon Tal were so far only found at distinct loci such as those with genetic fragility as indicated in Figures 4 f-g or enhancers without previous signatures of active DNA demethylation. We believe that the “increase” at around 600 bp distal to the CpG (or 3 kb upstream of caC excision, Figure 4c) is a sign of improper processing of the lesion due to PARP inhibition and a “spread” of the damage signature.

Figure 1 SSB signal distribution over the genome

SSB signal was counted over the whole mouse genome in 1000 bp bins and normalized to the signal in input samples. No genome-wide increase of SSBs in mESC treated with Talazoparib (Tal 1-3) is visible when compared to the control treatment (DMSO 1-3).

Reviewer #3 (Remarks to the Author)

In this study, Xu et al. report that poly(ADP-ribosylation) facilitates TET-TDG-mediated DNA demethylation via regulating base excision repair (BER) pathway. TDG is a DNA glycosylase and recognizes 5fC and 5csC. It mediates the removal of 5fC and 5csC from genomic DNA. Via BER pathway, unmodified cytosine is incorporated into genomic DNA to complete the active demethylation process. Here, the authors show that PARP1 PARylates a number of BER factors, such as TDG, for promoting dissociation of BER factors from genomic DNA upon repairing lesions. Overall, the manuscript provides an interesting hypothesis but lacks compelling evidence to support the major conclusions.

Specific points:

It is not surprising to see that many DNA damage repair factors are poly(ADP-ribosylated), particularly in an in vitro PARylation setting, as PARylation can be catalyzed on several amino acid residues. Numerous PARylation events occurs under extreme genotoxic stress conditions, and these events may be just passenger events. The key question that needs to be addressed is the biological function(s) of PARylation during DNA demethylation in mESCs. The authors used PARP inhibitor Talazoparib to examine the functions of PARylation, which might not be suitable, as this PARP inhibitor shows strong PARP1 trapping activities. Once PARP1 is trapped on genomic DNA, it will generate other secondary lesions, such as DNA double-strand breaks. Thus, it would be better to use PARP1 KO cells to examine the biological significance of PARP1-mediated PARylation.

Response: We appreciate the reviewer's comment. While it may not be surprising to see PARylation of BER proteins *in vitro*, most of the BER proteins examined have been identified in an *in vivo* screen for PARylated proteins in HeLa cells (Hendriks et al., 2019). However, this has not been independently validated. Therefore, we first needed to establish a highly controlled system to show that BER proteins are PARylated in the context BER and that their PARylation is specifically triggered by BER repair intermediates.

The reviewer recommends using PARP1 KO cells. This is an obvious proposal that we have carefully considered. However, genetic inactivation of PARP results in compensatory mechanisms via redundant functions or alternative pathways. Therefore, we decided to rather interfere with PARylation by temporal inhibition of PARylation activity than genetically deleting PARP.

The reviewer indicates that the use of PARP inhibitors may lead to PARP trapping, which can cause secondary lesions and possibly DSBs. In contrast to previous studies that investigated PARP-trapping, we use far lower concentrations of talazoparib and we did not induce any additional genotoxic stress such as damage by H₂O₂ or MMS. By using low PARP inhibitor concentrations, we inhibit PARP PARylation and avoid PARP trapping. This allows us to investigate the role of PARylation. Indeed, we can detect a very modest increase in DSBs in talazoparib treated ESCs (See Figure 3 below). This increase is far less pronounced compared to classical DNA damage inducers, which are used in PARP trapping experiments. We therefore don't consider PARP trapping as a strong factor in the described scenario. Importantly, talazoparib treatment of TDG wild type ESCs and TDG knock out ESCs results in the same very modest increase of DSBs. This shows that talazoparib affects mostly the repair of spontaneous DSBs in ESCs, but not TDG-BER related SSBs. Consistently, we detect an increase of SSB signal at sites prone to genetic instability, e.g. spontaneous SSBs.

To address the reviewer's important question of biological relevance of covalent BER protein PARylation, we performed additional experiments. We generated a PARylation deficient XRCC1 variant with 8 of 10 reported PARylation sites mutated (Hendriks et al., 2019) and studied its BER engagement in cells, and we made use of combinatorial treatments to control active DNA demethylation and de-PARylation. To first address the general relevance of PARylation in active DNA demethylation, we examined PAR levels in mESC in response to stimulation of active DNA demethylation by Vitamin C (Figure 1b). We also inhibited de-PARylation by treatment of cells with a PARG inhibitor (PDD 00017273) and found that this increases levels of fC and caC, consistent with an involvement of dynamic PARylation in TDG dependent active DNA demethylation (Figure 1c). As a direct examination of the role of PARylation in the dissociation of BER proteins in cells (without interfering with BER factor recruitment) is technical highly challenging (would require separation of function PARylation mutants), we investigated the re-association dynamics of PARylated BER proteins (TDG, XRCC1). When DNA demethylation was induced in ESCs by Vitamin C under conditions inhibiting de-PARylation by PARG inhibition, chromatin association of TDG and XRCC1 was reduced when compared to ESCs without PARGi treatment (Figure 2g). This shows that stabilizing PARylation interferes with BER protein binding to DNA. Finally, we could show that an XRCC1 variant, which is deficient in PARylation, can re-engage more efficiently in DNA repair in mESC than its fully PARylatable wildtype counterpart (Figure 3d-f)

Figure 2 DNA DSB upon Tal in wt and TDGnull mESC
Pulse field gel electrophoresis of wt and TDGnull ESCs treated with 5 nM Tal for the indicated time or 10 µg/ml Zeocin (Zeo) for 24 h. Data are part of another manuscript in preparation.

2. The working model proposed in this manuscript is that PARylation mediates the release of BER factors perhaps during intermediate steps, thus facilitates BER. However, it would be straightforward to examine if BER factors are retained at DNA lesion in PARP1 KO mESCs.

Response: This is a valid suggestion. We considered using *PARP1 KO mESCs* to investigate the role of PARylation in BER factor release. However, this is not as straightforward as it might seem. As PARylation of proteins in the vicinity of DNA strand-breaks is required for the initial recruitment of SSBR factors such as XRCC1 and LIG3 to damage, genetically removing PARP would thus also significantly affect the measurement of downstream repair steps including PARylation dependent dissociation of BER factors. Moreover, PARP1 and PARP2 would have to be removed both as they are redundant in DNA damage dependent PARylation. While this is possible, this is known to activate compensatory mechanisms that may interfere with the readout. Therefore, although we did carefully consider such experiments, we have decided against using *PARP1 KO mESCs*.

3. It may be strange to see that XRCC1 and POL beta bind to nicked DNA in the presence of PARP1 dead since PARP1 dead also binds to nicked DNA and may compete with XRCC1 and POL beta to interact with DNA.

Response: This is an important question. It is true that PARPdead may compete with XRCC1 and POL β for DNA binding. In our experimental setup we used a molar excess of POL β (~6x fold excess) and XRCC1 (~3.2 fold excess) over PARP1/PARP1dead. Our data clearly show that under the tested conditions the lower molar amount of PARPdead does not interfere with XRCC1 and POL β DNA binding.

4. The font in figures is too small.

The font in the figures was increased.

- Aleksandrov, R., Dotchev, A., Poser, I., Krastev, D., Georgiev, G., Panova, G., Babukov, Y., Danovski, G., Dyankova, T., Hubatsch, L., Ivanova, A., Atehin, A., Nedelcheva-Veleva, M. N., Hasse, S., Sarov, M., Buchholz, F., Hyman, A. A., Grill, S. W., & Stoynov, S. S. (2018). Protein Dynamics in Complex DNA Lesions. *Molecular Cell*, *69*(6), 1046-1061.e5. <https://doi.org/10.1016/j.molcel.2018.02.016>
- Antolin, A. A., Ameratunga, M., Banerji, U., Clarke, P. A., Workman, P., & Al-Lazikani, B. (2020). The kinase polypharmacology landscape of clinical PARP inhibitors. *Scientific Reports*, *10*(1), 2585. <https://doi.org/10.1038/s41598-020-59074-4>
- Baranello, L., Kouzine, F., Wojtowicz, D., Cui, K., Przytycka, T., Zhao, K., & Levens, D. (2014). DNA Break Mapping Reveals Topoisomerase II Activity Genome-Wide. *International Journal of Molecular Sciences*, *15*(7), 13111–13122. <https://doi.org/10.3390/ijms150713111>
- Ciccarone, F., Valentini, E., Zampieri, M., & Caiafa, P. (2015). 5mC-hydroxylase activity is influenced by the PARylation of TET1 enzyme. *Oncotarget*, *6*(27), 24333–24347. <https://doi.org/10.18632/oncotarget.4476>
- Cortázar, D., Kunz, C., Saito, Y., Steinacher, R., & Schär, P. (2007). The enigmatic thymine DNA glycosylase. *DNA Repair*, *6*(4), 489–504. <https://doi.org/10.1016/j.dnarep.2006.10.013>
- Cortazar, D., Kunz, C., Selfridge, J., Lettieri, T., Saito, Y., MacDougall, E., Wirz, A., Schuermann, D., Jacobs, A. L., Siegrist, F., Steinacher, R., Jiricny, J., Bird, A., Schar, P., Cortázar, D., Kunz, C., Selfridge, J., Lettieri, T., Saito, Y., ... Schär, P. (2011). Embryonic lethal phenotype reveals a function of TDG in maintaining epigenetic stability. *Nature*, *470*(7334), 419–423. <https://doi.org/10.1038/nature09672>
- Hardeland, U., Kunz, C., Focke, F., Szadkowski, M., & Schär, P. (2007). Cell cycle regulation as a mechanism for functional separation of the apparently redundant uracil DNA glycosylases TDG and UNG2. *Nucleic Acids Research*, *35*(11), 3859–3867. <https://doi.org/10.1093/nar/gkm337>
- Hendriks, I. A., Larsen, S. C., & Nielsen, M. L. (2019). An Advanced Strategy for Comprehensive Profiling of ADP-ribosylation Sites Using Mass Spectrometry-based Proteomics*. *Molecular & Cellular Proteomics*, *18*(5), 1010–1026. <https://doi.org/10.1074/mcp.TIR119.001315>
- Krüger, A., Bürkle, A., Hauser, K., & Mangerich, A. (2020). Real-time monitoring of PARP1-dependent PARylation by ATR-FTIR spectroscopy. *Nature Communications*, *11*(1), 2174. <https://doi.org/10.1038/s41467-020-15858-w>
- Ray Chaudhuri, A., & Nussenzweig, A. (2017). The multifaceted roles of PARP1 in DNA repair and chromatin remodelling. *Nature Reviews Molecular Cell Biology*, *18*(10), 610–621. <https://doi.org/10.1038/nrm.2017.53>
- Schomacher, L., Han, D., Musheev, M. U., Arab, K., Kienhöfer, S., von Seggern, A., & Niehrs, C. (2016). Neil DNA glycosylases promote substrate turnover by Tdg during DNA demethylation. *Nature Structural & Molecular Biology*, *23*(2), 116–124. <https://doi.org/10.1038/nsmb.3151>
- Steinacher, R., Barekati, Z., Botev, P., Kusnierczyk, A., Slupphaug, G., Schar, P., Kuśnierczyk, A., Slupphaug, G., & Schär, P. (2019). SUMOylation coordinates BERosome assembly in active DNA demethylation during cell differentiation. *The EMBO Journal*, *38*(1), e99242. <https://doi.org/10.15252/embj.201899242>
- Suskiewicz, M. J., Zobel, F., Ogden, T. E. H., Fontana, P., Ariza, A., Yang, J.-C., Zhu, K., Bracken, L., Hawthorne, W. J., Ahel, D., Neuhaus, D., & Ahel, I. (2020). HPF1 completes the PARP active site for DNA damage-induced ADP-ribosylation. *Nature*, *579*(7800), 598–602. <https://doi.org/10.1038/s41586-020-2013-6>
- Tolić, A., Ravichandran, M., Rajić, J., Đorđević, M., Đorđević, M., Dinić, S., Grdović, N., Jovanović, J. A., Mihailović, M., Nestorović, N., Jurkowski, T. P., Uskoković, A. S., & Vidaković, M. S. (2022). TET-

mediated DNA hydroxymethylation is negatively influenced by the PARP-dependent PARylation. *Epigenetics & Chromatin*, 15(1), 11. <https://doi.org/10.1186/s13072-022-00445-8>

Weber, A. R., Krawczyk, C., Robertson, A. B., Kuśnierczyk, A., Vågbo, C. B., Schuermann, D., Klungland, A., & Schär, P. (2016). Biochemical reconstitution of TET1–TDG–BER-dependent active DNA demethylation reveals a highly coordinated mechanism. *Nature Communications*, 7(0372), 10806. <https://doi.org/10.1038/ncomms10806>

Wu, W., Hill, S. E., Nathan, W. J., Paiano, J., Callen, E., Wang, D., Shinoda, K., van Wietmarschen, N., Colón-Mercado, J. M., Zong, D., De Pace, R., Shih, H.-Y., Coon, S., Parsadanian, M., Pavani, R., Hanzlikova, H., Park, S., Jung, S. K., McHugh, P. J., ... Nussenzweig, A. (2021). Neuronal enhancers are hotspots for DNA single-strand break repair. *Nature*. <https://doi.org/10.1038/s41586-021-03468-5>

REVIEWER COMMENTS

Reviewer #1 (Remarks to the Author):

In response to DNA damage, a subset of the poly(ADP-ribose) polymerase (PARP) family members, most notably PARP1, become activated and synthesize chains of ADP-ribose that are covalently linked either to the PARP enzyme itself or adjacent proteins, including histones and DNA repair proteins. While it is well documented that poly (ADP-ribosylated) PARP molecules in the vicinity of the DNA image serve as the signal to recruit DNA repair enzyme, much less is known about the effect of adding poly(ADP-ribose) chains to DNA repair enzymes. In this revised manuscript, the authors examine the examine a specialized form of BER that is involved in DNA demethylation. The authors have been very responsive to the previous critiques and have provided additional data in support of their model in which poly (ADP-ribosylation) of BER enzymes increases the efficiency of the repair reaction by promoting the dissociation of the enzymes from DNA. The role of PARP1 in BER has remained controversial as in the prevailing passing the baton model the BER intermediates are bound in product complexes and so may be sequestered away from PARP1. Here the authors provide convincing evidence that PARP1 is activated by both AP site and SSB intermediates generated during BER-mediated DNA demethylation. More importantly, they show that the efficiency of the BER reaction is increased by apol(ADP-ribosylation) of the BER enzymes which promotes their dissociation of DNA. Overall, while the changes reported are relatively small (eg Fig. 1c)and some instances, the protein ADP-ribosylation is not very convincing (eg Lig3, Fig. 2c), the results together do provide compelling evidence that poly(ADP-ribosylation) of the repair enzymes does enhance the repair reaction. Thus, PARP1 is not required for BER but its auto ADP ribosylation enhances repair enzyme recruitment and its ADP-ribosylation of BER enzymes directly enhances the repair reaction.

Minor comments.

Abstract, first sentence needs to specify which PARP enzymes are being referred to.
pg. 2 last sentence shown be " to themselves and..."

Reviewer #2 (Remarks to the Author):

I would like to thank the authors for addressing most of the raised remarks in their revised manuscript.

Minor comments:

Reference #2 is missing

Reviewer #3 (Remarks to the Author):

In mammalian cells, DNA methylation predominantly occurs within CG di-nucleotides of double-stranded DNA, following a symmetric pattern. Considering the possibility of TDG mediating glycosylation of methylated cytosine for DNA demethylation, a potential outcome could be the generation of adjacent AP sites symmetrically. This, in turn, might lead to the immediate formation of DSBs rather than SSBs. Consequently, if TDG indeed facilitates DNA methylation, DSB repair, rather than SSB repair could emerge as a significant player. This line of reasoning raises questions about the validity of the major conclusions presented in the manuscript. Furthermore, it appears that the concerns I previously raised have not been adequately addressed by the authors. It's worth noting that talazoparib exhibits profound toxicity towards mammalian cells. Even at very low concentrations, talazoparib treatment is able to induce PARP1/2 trapping, resulting in the initiation of DSBs through the response of replication stress.

Reviewer #2 comments on Reviewer #3 comments:

I agree with Reviewer #3 that although TDG mediated demethylation can also occur at non CG sites (deNizio et al, J. Mol. Biol, 2021), the majority of TDG activity will likely be at CG di-nucleotides. TDG then could indeed lead to the direct formation of DSBs due to the close proximity of the AP sites generated. Immediately after replication hemi-methylated sites are present at CG sites, because the C in the newly synthesized strand has not yet been methylated. These hemi-methylated sites are recognized by the maintenance DNA methyltransferase 1 (DNMT1) which then mediates the maintenance of methylation marks by methylating the C in the new strand. If TDG would act before DNMT1, then still SSBs and not DBSs would be formed. There have been reports stating that TDG is actively degraded in S-phase, so then this scenario would not be very likely (Hardeland, NAR, 2007). I am not aware whether this question has been addressed in more detail in the literature. Concerning Talazoparib, I also agree that it can be toxic to cells, even if cells are proficient in HR and will result in S-phase specific formation of DSBs (Michelena, Nat. Comm., 2018). The authors did however use relatively low concentration of Talazoparib. It would be better to use PARP1 knockout cells. The authors have argued that they preferred to not use PARP1 knockout cells because of compensatory mechanisms and have instead included experiments using PARG inhibition. Instead of using PARP1 knockout cells, one could still induce short-term loss of PARP1 through siRNA or shRNA. In addition a non-trapping but potent PARP1 Inhibitor, like Veliparib could be used.

RESPONSE TO REVIEWERS' COMMENTS

We would like to thank Reviewers #1 and #2 for their valuable input and approval of our revised manuscript. We specified the PARP enzymes in the abstract and corrected the typos and error in the reference accordingly.

We also appreciate the comments of Reviewer #3. The points raised are relevant and address very fundamental questions of coordination of TET-TDG-BER-mediated active DNA demethylation and the role of PARP and PARylation in this essential process. We have been investigating these and related mechanistic questions thoroughly over that past years and would like to take the opportunity to use some of the published and unpublished experimental evidence to address the reviewers' comments and clarify our current view on the topic.

Comment Reviewer #3: *"In mammalian cells, DNA methylation predominantly occurs within CG dinucleotides of double-stranded DNA, following a symmetric pattern. Considering the possibility of TDG mediating glycosylation of methylated cytosine for DNA demethylation, a potential outcome could be the generation of adjacent AP sites symmetrically. This, in turn, might lead to the immediate formation of DSBs rather than SSBs. Consequently, if TDG indeed facilitates DNA methylation, DSB repair, rather than SSB repair could emerge as a significant player. This line of reasoning raises questions about the validity of the major conclusions presented in the manuscript. Furthermore, it appears that the concerns I previously raised have not been adequately addressed by the authors. It's worth noting that talazoparib exhibits profound toxicity towards mammalian cells. Even at very low concentrations, talazoparib treatment is able to induce PARP1/2 trapping, resulting in the initiation of DSBs through the response of replication stress."*

Our response:

We thank the Reviewer for bringing up these important points. To avoid a possible misunderstanding we would first like to note that TDG does not glycosylate methylated cytosines (mC), but excises the oxidized mC derivatives fC and caC generated by TET hydroxylases from the DNA. It does so by cleaving the N-glycosidic bond of the fC and caC nucleotides, thereby producing so-called abasic-sites (AP-site) in DNA^{1,2}, which are then processed to DNA single strand-breaks (SSB) and repaired by SSB-repair (SSBR). BER and hence also active DNA demethylation is a process where DNA base excision by a DNA glycosylase is followed by SSBR. This mode of TET-TDG-BER mediated active DNA demethylation is well-studied and accepted³⁻¹⁰.

How base excision by TDG is coordinated at symmetrically modified CpGs so that DNA double strand-break (DSB) formation can be avoided is indeed an important question. In this regard, it was shown previously that TDG action is very tightly controlled by different molecular mechanisms^{11,12}. One mechanism particularly relevant for DSB formation is the rate-limiting and regulated dissociation of the glycosylase from the AP-site once it has cleaved the base. The regulation of this step requires posttranslational SUMO modification of TDG and/or the association of downstream acting BER enzymes, like APE1, guaranteeing the coupling of DNA based excision by TDG with the repair of the AP-site^{7,11,13}. We demonstrated previously that the TET-TDG-BER mechanism can indeed demethylate both DNA strands of symmetrically methylated CpGs. Importantly, however, these biochemical experiments also showed that once base excision is initiated in one strand, BER is completed before the DNA demethylation of the other strand can start. These observations thus show that demethylation of symmetrically methylated CpGs occurs in a sequential manner, thereby generally avoiding DSB formation⁹.

Another relevant and well-established mechanism of TDG control concerns its cell cycle regulation. We and others have demonstrated that TDG undergoes proteasome-mediated degradation at the entry into S-phase and then is kept at undetectably low levels during the entire S-phase¹⁴⁻¹⁶. Thus, as Reviewer #2 correctly pointed out in response to the comments of Reviewer #3, it is very unlikely that TDG-BER will initiate SSBs or DSBs during DNA replication, in an “unchallenged” condition. We cannot exclude, however, that under conditions of BER saturation/inhibition, some unrepaired SSBs generated before S-phase will be converted to DSB at replication forks during S-phase. Therefore, based on the experimental evidence already available, TDG dependent BER is associated with the formation of SSBs, but not with the formation of DSBs, and we consider the conclusions drawn in our manuscript valid and justified.

Reviewer #3 is correct in stating that drug-induced PARP-trapping can cause cytotoxicity. This, however, depends on the drug, its dosage and the experimental setup applied. PARP-trapping will occur under conditions of PARP inhibition and this is best detectable when cells are treated with DNA damaging agents (e.g. MMS) and higher concentrations of inhibitor^{17,18}. For Talazoparib (Tal), it was shown that concentrations above 0.1 μM do induce detectable PARP-trapping whereas concentrations below 0.1 μM do not, even in the presence of induced DNA damage¹⁹. Notably, in our cell culture experiments, we used 20-fold lower Tal concentrations (5 nM) and no exogenous DNA damage induction at all. Under these conditions, we were not able to detect an increase in PARP1 trapping (chromatin association) (Figure 1, new Supplementary Fig. 4a).

Figure 1 Chromatin association of PARP1 upon Talazoparib

Top: Representative immunoblot image of fractionated ESCs upon 5 nM Tal (and 0.01% MMS) probed for PARP1 and Tubulin. Chr: chromatin bound, cyt: cytoplasmic, nuc: nuclear soluble. Bottom: Quantification of chromatin bound or soluble PARP1. Depicted is the mean +/- SD of n=3. *: p<0.05

As communicated before, we nevertheless tested potential DSB formation by Tal (5 nM) in our mouse ESC system (in the context of another study; manuscript in preparation). We did not find an increase of DSBs upon depletion of TDG and only a marginal increase after Tal treatment. Importantly however, this increase was equally high in TDG proficient (TDGwt) and deficient ESC cells (TDGnull) (Figure 2). This shows that in our experimental setup, PARP inhibition by Tal does

induce a small amount of DSBs but that these DSBs are generated by a pathway independent of TDG. We therefore conclude that treatment of ESCs with 5 nM Talalzoparib does not convert TDG induced DNA SSBs to detectable DSBs.

Figure 2 DNA DSB upon Tal in wt and TDGnull mESC

Pulse field gel electrophoresis of wt and TDGnull ESCs treated with 5 nM Tal for the indicated time or 10 µg/ml Zeocin (Zeo) for 24 h. Data are part of another manuscript in preparation.

Reviewer #3 also refers to “previously raised concerns”, which we assume are concerns about a bias in the DNA-SSB-seq analysis due to the potential conversion of SSBs to DSBs following Tal treatment. As mentioned before, we agree that this is an important point. Following up on our above outlined evidence and argumentation against a dominant role of TDG dependent DSB formation upon PARP inhibition, we would like to add the following considerations. If DSB formation by PARP1-trapping were a dominant mechanism in our experimental setup and if DSB were efficiently detected by our method of SSBs analysis, we would expect to see an increase of sites with a significant enrichment of DNA breaks (peaks) across the genome upon Tal treatment. What we do see, however, is a reduction in the number of SSB peaks to 54% of the level observed in untreated ESCs (28’500 to 15’500). If this reduction of SSBs was due to their conversion to DSBs, we would expect to detect a significant TDG-dependent increase in DSBs (Figure 2), which we don’t. Moreover, our SSB-seq data show that Tal treatment affects SSB formation differently at sites of active DNA demethylation and elsewhere, supporting a specific and direct mechanistic association of PARP-activity with TET/TDG-mediated active DNA demethylation but not excluding TET/TDG independent effect at other sites. Based on this data-supported consideration, we argue that the reduction of SSBs upon Tal treatment of ESCs is best explained by an interruption of turnover of TDG-dependent BER (and thus SSB formation) and that the conversion of TDG-generated SSBs to DSBs may occasionally occur but represents a minor event.

Reviewer #3 mentions the possible use of PARP1 knockout cells and Reviewer #2 suggests using knockdown cells or another PARP inhibitor (Veliparib) as an alternative or validating approach. We have taken these suggestions very seriously and would like to explain our reasons not to do so in more detail. After very careful consideration of the evidence available (published and unpublished), we concluded that experiments using PARP depleted ESCs would not be suitable to investigate the mechanism addressed in our manuscript. On the one hand, there is the issue of difficult to control compensatory mechanisms following PARP depletion (PARP1 and/or PARP2), including the activation and/or deregulation of alternative DNA repair mechanisms (e.g. HR, NER²⁰) or protein activities (e.g.

TET activity itself²¹) as well documented. On the other hand, and more specifically, the absence of PARP1 (and PARP2) and PARylation of histones will abolish the recruitment of BER/SSBR factors through non-covalent interactions with PAR, as shown before²². Notably, depletion of PARP1 or PARP2 separately is not sufficient to affect recruitment of XRCC1 to DNA damage significantly. Combined PARP1&2 depletion, however, does abolish XRCC1 recruitment to damaged chromatin, showing that PARP1 can compensate for the absence of PARP2 and vice versa. So, PARP1&2 depletion would be an option to affect BER, but this would perturb the dynamics of BER already at the stage of BER complex assembly and, hence, not allow the downstream steps of turnover/dissociation of engaged BER proteins to be investigated. Or in other words, the perturbation of the upstream role of PARP1 in BER complex assembly would prevent the investigation of its downstream role of BER complex turnover by covalent modification of its components. In our opinion, such an experiment would produce inconclusive results.

Likewise, we also considered the possibility to apply and to some extent also applied different PARP inhibitors. However, as pharmacological PARP inhibition, including by Tal and Veliparib^{23,24}, was generally shown to affect the recruitment of the BER/SSBR machinery (XRCC1), such experiments would not provide further insight into the function of BER protein PARylation downstream of repair initiation and the associated dynamics of SSB appearance. Therefore, after careful consideration, we decided to use a different approach using PARG inhibition (PARGi) and the generation of a PAR-deficient XRCC1 variant to gain deeper insight into the biology of the regulatory role of PARylation in active DNA demethylation. The advantage of using PARGi over using PARP-depleted cells, is that the initial non-covalent PAR-mediated recruitment and subsequent covalent PARylation of BER factors is not affected. Instead, de-PARylation of BER proteins is prevented and the proteins remain hyper-PARylated. This, together with the use of PARylation defective XRCC1 allowed us to more precisely study the role of covalent PARylation of BER proteins and from different angles, i.e., the failure of PARylation of the central BER/SSBR factor XRCC1 and the behavior of the same protein when it is hyper-PARylated. This enabled us to show that PARylation of BER proteins results in reduced DNA binding and altered BER mediated DNA demethylation, in line with the biochemical evidence in our manuscript.

We would like to state, that the Reviewers' questions guided us in significantly improving our manuscript in a major revision. We added new experiments and data that substantiated and validated our biochemical findings on the function of BER/SSBR protein PARylation, providing important novel insight into the molecular mechanism of BER and active DNA demethylation in cells. Hence, using different independent experimental cell biological and biochemical approaches, our data establish the biological significance and molecular function of covalent BER protein PARylation in the regulation of the multienzyme actions during active DNA demethylation, a mechanistic concept that may apply to other multienzyme protein machines beyond DNA repair and demethylation.

In response to the Reviewer #3's concern, we added additional experimental evidence to supplementary data (New Supplementary Figure 4a), showing that treatment of mESC with 5 nM Talazoparib does not trap PARP1 to chromatin in the absence of induced DNA damage.

1. Maiti, A. & Drohat, A. C. Thymine DNA glycosylase can rapidly excise 5-formylcytosine and 5-carboxylcytosine: potential implications for active demethylation of CpG sites. *J. Biol. Chem.* **286**, 35334–8 (2011).
2. Jacobs, A. L. & Schär, P. DNA glycosylases: in DNA repair and beyond. *Chromosoma* **121**, 1–20

- (2011).
3. DeNizio, J. E. *et al.* TET-TDG active DNA demethylation at CpG and non-CpG sites. *J. Mol. Biol.* 166877 (2021) doi:<https://doi.org/10.1016/j.jmb.2021.166877>.
 4. Song, C. X. *et al.* Genome-wide profiling of 5-formylcytosine reveals its roles in epigenetic priming. *Cell* **153**, 678–691 (2013).
 5. Wang, D. *et al.* Active DNA demethylation promotes cell fate specification and the DNA damage response. *Science (80-.)*. **378**, 983–989 (2022).
 6. Shen, L. *et al.* Genome-wide Analysis Reveals TET- and TDG-Dependent 5-Methylcytosine Oxidation Dynamics. *Cell* **153**, 692–706 (2013).
 7. Steinacher, R. *et al.* SUMOylation coordinates BERosome assembly in active DNA demethylation during cell differentiation. *EMBO J.* **38**, e99242 (2019).
 8. Onodera, A. *et al.* Roles of TET and TDG in DNA demethylation in proliferating and non-proliferating immune cells. *Genome Biol.* **22**, 186 (2021).
 9. Weber, A. R. *et al.* Biochemical reconstitution of TET1–TDG–BER-dependent active DNA demethylation reveals a highly coordinated mechanism. *Nat. Commun.* **7**, 10806 (2016).
 10. Wu, W. *et al.* Neuronal enhancers are hotspots for DNA single-strand break repair. *Nature* (2021) doi:[10.1038/s41586-021-03468-5](https://doi.org/10.1038/s41586-021-03468-5).
 11. Hardeland, U., Steinacher, R., Jiricny, J. & Schär, P. Modification of the human thymine-DNA glycosylase by ubiquitin-like proteins facilitates enzymatic turnover. *EMBO J.* **21**, 1456–1464 (2002).
 12. Waters, T. R., Gallinari, P., Jiricny, J. & Swann, P. F. Human Thymine DNA Glycosylase Binds to Apurinic Sites in DNA but Is Displaced by Human Apurinic Endonuclease 1. *J. Biol. Chem.* **274**, 67–74 (1999).
 13. Steinacher, R. & Schär, P. Functionality of human thymine DNA glycosylase requires SUMO-regulated changes in protein conformation. *Curr. Biol.* **15**, 616–623 (2005).
 14. Hardeland, U., Kunz, C., Focke, F., Szadkowski, M. & Schär, P. Cell cycle regulation as a mechanism for functional separation of the apparently redundant uracil DNA glycosylases TDG and UNG2. *Nucleic Acids Res.* **35**, 3859–3867 (2007).
 15. Aranda, S. *et al.* Thymine DNA glycosylase regulates cell-cycle-driven p53 transcriptional control in pluripotent cells. *Mol. Cell* (2023) doi:[10.1016/j.molcel.2023.07.003](https://doi.org/10.1016/j.molcel.2023.07.003).
 16. Slenn, T. J. *et al.* Thymine DNA Glycosylase is a CRL4Cdt2 Substrate. *J. Biol. Chem.* (2014) doi:[10.1074/jbc.M114.574194](https://doi.org/10.1074/jbc.M114.574194).
 17. Murai, J. *et al.* Rationale for poly(ADP-ribose) polymerase (PARP) inhibitors in combination therapy with camptothecins or temozolomide based on PARP trapping versus catalytic inhibition. *J Pharmacol Exp Ther* **349**, 408–416 (2014).
 18. Michelena, J. *et al.* Analysis of PARP inhibitor toxicity by multidimensional fluorescence microscopy reveals mechanisms of sensitivity and resistance. *Nat. Commun.* **9**, 2678 (2018).
 19. Mahadevan, J. *et al.* Dynamics of endogenous PARP1 and PARP2 during DNA damage revealed by live-cell single-molecule imaging. *iScience* **26**, 105779 (2023).
 20. Ray Chaudhuri, A. & Nussenzweig, A. The multifaceted roles of PARP1 in DNA repair and chromatin remodelling. *Nat. Rev. Mol. Cell Biol.* **18**, 610–621 (2017).

21. Tolić, A. *et al.* TET-mediated DNA hydroxymethylation is negatively influenced by the PARP-dependent PARylation. *Epigenetics Chromatin* **15**, 11 (2022).
22. Hanzlikova, H., Gittens, W., Krejcikova, K., Zeng, Z. & Caldecott, K. W. Overlapping roles for PARP1 and PARP2 in the recruitment of endogenous XRCC1 and PNKP into oxidized chromatin. *Nucleic Acids Res.* **45**, gkw1246 (2016).
23. Zhao, M.-L. *et al.* Temporal recruitment of base excision DNA repair factors in living cells in response to different micro-irradiation DNA damage protocols. *DNA Repair (Amst)*. **126**, 103486 (2023).
24. Horton, J. K. *et al.* XRCC1-mediated repair of strand breaks independent of PNKP binding. *DNA Repair (Amst)*. **60**, 52–63 (2017).